



# Modeling of land-surface interactions in the PALM model system 6.0: Land surface model description, first evaluation, and sensitivity to model parameters

Katrin Frieda Gehrke[1], Matthias Sühring[1], and Björn Maronga[1,2]

[1]Leibniz University Hannover, Institute of Meteorology and Climatology, Hannover, Germany
[2]University of Bergen, Geophysical Institute, Bergen, Norway

**Correspondence:** Katrin Gehrke (gehrke@muk.uni-hannover.de)

**Abstract.** In this paper the land-surface model embedded in the PALM model system is described and evaluated against in-situ measurement data in Cabauw. For this, two consecutive clear-sky days are simulated and the components of surface energy balance, as well as near-surface potential temperature, humidity and horizontal wind speed are compared against observation data. For the simulated period, components of the energy balance agree well during day- and nighttime, and also the daytime
Bowen ratio agrees fairly well compared to the observations. Although the model simulates a significantly more stably-stratified nocturnal boundary layer compared to the observation, near-surface potential temperature and humidity agree fairly well during day. Moreover, we performed a sensitivity study in order to investigate how much the model results depend on land-surface and soil specifications, as well as atmospheric initial conditions. By this, we find that a false estimation of the leaf area index, the albedo, or the initial humidity causes a serious misrepresentation of the daytime turbulent sensible and latent heat fluxes. During night, the boundary-layer characteristics are mostly affected by grid size, surface roughness, and the applied radiation
schemes.

## 1 Introduction

The land surface influences atmospheric dynamics significantly through the exchange of energy, mass, and momentum. Therefore, an accurate representation of surface-atmosphere interactions is essential for any numerical modeling of the atmospheric
boundary layer (Garratt, 1993; Betts et al., 1996). More specifically, surface roughness as well as sensible and latent heat fluxes at the surface act as the lower boundary condition for the momentum, temperature and humidity equations in the atmosphere, respectively. If this information is not available, it is necessary to parameterize the land-surface processes with a Land-Surface Model (LSM). In simple terms, the input to an LSM is the type of surface, vegetation and soil, as well as the radiative forcing. Based on that, LSMs solve the surface energy budget equation by means of a set of prognostic equations for the surface tem-
perature and compute soil moisture and temperature in a multi-layer soil model. Nevertheless, the results strongly depend on the input data (e.g. Avissar and Pielke, 1989).

LSMs are required in various situations: Most often, observational data of the sensible and latent heat fluxes are unavailable or do not adequately reflect the complexity of the landscape. E.g. in case of forecasts, it is trivial that a prognostic approach has



to be chosen. Moreover, the use of an LSM allows for an interaction between atmosphere and surface on shortest timescales
and at every point in space, whereas the prescription of fluxes cannot represent this interaction. This plays an important role
in simulating feedback effects, e.g. between mesoscale circulations and underlying heterogeneous surfaces (e.g. Patton et al.,
2005; Huang and Margulis, 2010), between clouds and radiation (Lohou and Patton, 2014; Horn et al., 2015), or in cities,
where shadows of buildings are another reason for highly heterogeneous heat fluxes.

The PALM model system (Maronga et al., 2015, 2020) has been applied for studying a variety of atmospheric boundary-
layer flows for about 20 years. Since 2015, PALM comes with a fully interactive LSM. Originally, it was designed similar to
the LSM in the LES model DALES (Heus et al., 2010), which contains parameterizations of the Tiled ECMWF Scheme for
Surface Exchanges over Land (TESSEL/HTESSEL, Viterbo and Beljaars, 1995; van den Hurk et al., 2000; Balsamo et al.,
2009, 2011), but also some found in the *Interaction Sol-Biosphère-Atmosphère* model (ISBA, Noilhan and Mahfouf, 1996) and
own extensions. The LSM in PALM has first been described by Maronga and Bosveld (2017) and has been used with respect
to radiation fog by Maronga and Bosveld (2017) and Schwenkel and Maronga (2019), also including Cabauw (Netherlands)
data for evaluation. Recently, it has also been employed in urban environments (see first results in Maronga et al., 2020).

Other coupled LES-LSM implementations are found in e.g. UCLA-LES (Huang et al., 2011), ICON-LEM (Dipankar et al.,
2015), or the WRF model (Skamarock et al., 2019) in LES mode coupled to NOAH-LSM (Chen and Dudhia, 2001). In
early literature, the focus of coupled LES-LSM studies was mainly to analyze the feedback effect that creates heterogeneity
at the surface (e.g. Patton et al., 2005; Huang et al., 2009; Brunsell et al., 2011). Today, more and more studies require the
coupled LES-LSM approach to simulated realistic cases e.g. in the urban environment or for wind turbine applications. As the
methodology gains foothold in engineering and industry, it becomes increasingly important that the embedded land-surface
representation in PALM reflects reality (Maronga et al., 2020).

This paper is part of a series featuring different parts of the PALM model system 6.0 in this special issue. For the user, a
systematic sensitivity test of relevant land surface parameters with the LSM in PALM is of particular interest. In the present
study, we evaluate the LSM embedded into PALM against Cabauw data for a selected period with clear-sky conditions and
small large-scale advection. By this we ensure as far as possible that the developing boundary-layer is not affected by non-local
processes, which we neglect in the simulations of this study. Moreover, we will determine key parameters that influence the
diurnal cycle in a coupled LES-LSM framework. The model sensitivity will be analyzed by means of a comprehensive set of
simulations varying land-surface and soil parameters individually. Therewith, the present study complements the earlier work
of Maronga and Bosveld (2017), who focused on the nocturnal boundary layer with developing radiation fog.

Section 2 of the paper describes physical and technical aspects of the LSM in PALM. Section 3 provides information about
the Cabauw Experimental Site for Atmospheric Research (CESAR Monna and Bosveld, 2013) and the observations used. In
Sect. 4 the model setup and initialization is described and a complete list of the conducted simulations is given. Section 5
outlines the results of the sensitivity study. The validity and limitations of the LSM is discussed. Finally, the summary and
conclusions are drawn in Sect. 6 and an outlook is given.



## 2 Description of the land-surface model (LSM) in PALM

The LSM implemented in PALM consists of a solver for the energy balance of the Earth's surface in combination with a multi-layer soil scheme. The original scheme was designed for vegetated surfaces and bare soil only, but it has been adapted for paved

surface materials like asphalt and concrete, and a simplified version for inland and sea water surfaces has been added, too. A tile approach is available for vegetation surfaces, in which the surface can comprise of a fraction of bare soil and a fraction covered with vegetation. Furthermore, the LSM has a liquid water reservoir on plants and soil to store and evaporate liquid water from precipitation interception and dew formation. A liquid water reservoir is also available when the surface type is set to pavement, representing the ability of impervious surfaces to store a limited amount of precipitation water on the surface.

The specific implementation of PALM's LSM, which is derived from the HTESSEL scheme is described below. For further details, see also Viterbo and Beljaars (1995); van den Hurk et al. (2000); Balsamo et al. (2009, 2011) and literature referenced therein.

### 2.1 Energy balance solver

Within the LSM, the energy balance is calculated as

$$\frac{dT_0}{dt} C_0 = R_{\mathrm{n}} - H - LE - G \,, \tag{1}$$

where $C_0$ and $T_0$ are the heat capacity and radiative temperature of the surface, respectively. Note that $C_0$ is zero by default in case of surfaces covered by vegetation or water surfaces, where it is assumed that $T_0$ is the temperature of a skin layer covering the surface that does not have a heat capacity. In all other cases (i.e., pavements and bare soils), no skin layer is assumed (see below). $R_{\mathrm{n}}$, $H$, $LE$, and $G$ are the net radiation, sensible heat flux, latent heat flux, and ground (soil) heat flux at the surface,

respectively. $R_{\mathrm{n}}$ is defined positive downwards whereas $H$, $LE$, and $G$ are defined positive away from the surface. $R_{\mathrm{n}}$ is defined through the sum of the radiative fluxes:

$$R_{\mathrm{n}} = SW_{\downarrow} - SW_{\uparrow} + LW_{\downarrow} - LW_{\uparrow} \,, \tag{2}$$

where $SW_{\downarrow}$, $SW_{\uparrow}$, $LW_{\downarrow}$, and $LW_{\uparrow}$ are the shortwave incoming (downward), shortwave outgoing (upward), longwave incoming (downward), and longwave outgoing (upward) flux, respectively. The radiation components are defined positive according to

their direction ($SW_{\downarrow}$ and $LW_{\downarrow}$ positive downwards; $SW_{\uparrow}$ and $LW_{\uparrow}$ positive upwards). The radiative fluxes are provided by one of the available radiation schemes in PALM (for details, see Maronga et al., 2020).

### 2.1.1 Parameterization of fluxes

The turbulent heat fluxes both are parameterized using a resistance parameterization. $H$ is calculated as

$$H = -\rho \, c_p \frac{1}{r_{\mathrm{a}}} (\theta_{\mathrm{mo}} - \theta_0) \,, \tag{3}$$

where $\rho$ is the density of the air, $c_p = 1005 \, \mathrm{J \, kg^{-1} \, K^{-1}}$ is the specific heat at constant pressure, and $r_{\mathrm{a}}$ is the aerodynamic resistance. $\theta_0$ and $\theta_{\mathrm{mo}}$ are the potential temperature at the surface and at a fixed height within the atmospheric surface layer





(at height $z_{\mathrm{mo}}$, usually at height of the first atmospheric grid level, i.e., $z_{\mathrm{mo}} = 0.5\Delta_z$ where $\Delta_z$ is the vertical grid spacing), respectively. The potential temperature is linked to the actual temperature via the Exner function, viz.

$$\Pi = \left(\frac{p}{1000\,\mathrm{hPa}}\right)^{R_{\mathrm{d}}/c_p} , \tag{4}$$

with pressure $p$ and the gas constant for dry air $R_{\mathrm{d}}$. $r_a$ is calculated via Monin-Obukhov similarity theory (MOST) as

$$r_{\mathrm{a}} = \frac{\theta_{\mathrm{mo}} - \theta_0}{u_* \, \theta_*} , \tag{5}$$

where $u_*$ and $\theta_*$ are the friction velocity and characteristic temperature scale, respectively, and which are calculated locally based on MOST. Note that the values for $u_*$ and $\theta_*$ from the previous time step are used in Eq. (5). The roughness lengths are individually set for momentum, heat, and moisture (see Table 1). Note that for water surfaces, a Charnock parameterization

can be switched on for taking into account the effect of subgrid-scale wave motions through the roughness lengths. For details on the particular implementation of MOST and the Charnock parameterization in PALM, see Maronga et al. (2020). Note that $r_{\mathrm{a}}$ is calculated based on $u_*$ and $\theta_*$ values from the current time step to calculate $H$ at the prognostic time step.

The ground heat flux, $G$, is parameterized after (Duynkerke, 1999) as

$$G = \Lambda(T_0 - T_{\mathrm{soil},1}), \tag{6}$$

with $\Lambda$ being the total thermal conductivity between skin layer and the uppermost soil layer. $T_{\mathrm{soil},1}$ is the temperature of the uppermost soil layer (calculated at the center of the layer). $\Lambda$ is calculated via a resistance approach as a combination of the conductivity between the canopy and the soil-top ($\Lambda_{\mathrm{skin}}$, constant value) and the conductivity of the top half of the uppermost soil layer ($\Lambda_{\mathrm{soil}}$):

$$\Lambda = \frac{\Lambda_{\mathrm{skin}}\Lambda_{\mathrm{soil}}}{\Lambda_{\mathrm{skin}} + \Lambda_{\mathrm{soil}}} . \tag{7}$$

When no skin layer is used (i.e. in case of pavements and bare soils), $\Lambda$ reduces to the heat conductivity of the uppermost soil layer, viz.

$$\Lambda = \frac{\lambda_{\mathrm{T,pave}}}{\Delta z_{\mathrm{soil},1}} , \tag{8}$$

with $\lambda_{\mathrm{T,pave}}$ being the thermal conductivity of the pavement and $\Delta z_{\mathrm{soil},1}$ being the depth of the uppermost soil layer. In this case, it is assumed that the soil temperature is constant within the uppermost 25 % of the top soil layer and equals the radiative

temperature at the surface. $C_0$ is then set to a non-zero value according to the material properties and the layer thickness.

The total latent heat flux, $LE$, is parameterized as

$$LE = -\rho \, l_{\mathrm{v}} \, \frac{1}{r_{\mathrm{a}} + r_{\mathrm{s}}} (q_{\mathrm{v,mo}} - q_{\mathrm{v,sat}}(T_0)) . \tag{9}$$

Here, $l_{\mathrm{v}} = 2.5 \times 10^6 \, \mathrm{J\,kg^{-1}}$ is the latent heat of vaporization, $r_{\mathrm{s}}$ is the total surface resistance, $q_{\mathrm{mo}}$ is the water vapor mixing ratio at height $z_{\mathrm{mo}}$, and $q_{\mathrm{sat}}$ is the water vapor mixing ratio at saturation at the surface, which is a function of $T_0$. In practice,



up to three individual components are calculated for vegetated surfaces. Transpiration of the the vegetated fraction ($LE_\mathrm{veg}$) is parameterized as

$$LE_\mathrm{veg} = -\rho\, l_\mathrm{v}\, \frac{1}{r_\mathrm{a} + r_\mathrm{c}} (q_\mathrm{v,mo} - q_\mathrm{v,sat}(T_0))\,, \tag{10}$$

where $r_\mathrm{c}$ is the canopy resistance. Analogous the bare soil fraction evaporation ($LE_\mathrm{soil}$) is calculated via

$$LE_\mathrm{soil} = -\rho\, l_\mathrm{v}\, \frac{1}{r_\mathrm{a} + r_\mathrm{soil}} (q_\mathrm{v,mo} - q_\mathrm{v,sat}(T_0))\,, \tag{11}$$

with $r_\mathrm{soil}$ being the soil resistance. The liquid water reservoir evaporation ($LE_\mathrm{liq}$) is given by

$$LE_\mathrm{liq} = -\rho\, l_\mathrm{v}\, \frac{1}{r_\mathrm{a}} (q_\mathrm{v,mo} - q_\mathrm{v,sat}(T_0))\,, \tag{12}$$

i.e., only the aerodynamic resistance exists for liquid water. The total evapotranspiration is then given by a combination of the three individual components by (see Viterbo and Beljaars, 1995)

$$LE = c_\mathrm{veg}(1 - c_\mathrm{liq})LE_\mathrm{veg} + c_\mathrm{liq}LE_\mathrm{liq} + (1 - c_\mathrm{veg})(1 - c_\mathrm{liq})LE_\mathrm{soil}\,. \tag{13}$$

Here, $c_\mathrm{veg}$ and $c_\mathrm{liq}$ are the fractions of the surface covered with vegetation and liquid water, respectively. Liquid water from precipitation can be stored on the vegetation and bare soil. Note that for paved and water surfaces both $LE_\mathrm{veg}$ and $LE_\mathrm{soil}$ are set to zero and the only possible source of evaporation is the liquid water reservoir.

All equations above are solved locally for each surface element of the model grid.

### 2.1.2   Liquid water reservoir

In order to account for the evaporation of liquid water on plants on impervious surfaces, an additional equation is solved for the liquid water reservoir:

$$\frac{d\,m_\mathrm{liq}}{d\,t} = \frac{LE_\mathrm{liq}}{\rho_l\, l_\mathrm{v}}\,, \tag{14}$$

where $m_\mathrm{liq}$ and $\rho_l$ are the water column on the surface and the density of water, respectively. The maximum amount of water that can be stored on plants is calculated via

$$m_\mathrm{liq,max} = \min\left(1, m_\mathrm{liq,ref} \cdot c_\mathrm{veg} \cdot LAI + (1 - c_\mathrm{veg})\right)\,, \tag{15}$$

where $m_\mathrm{liq,ref} = 0.2\,\mathrm{mm}$ is the reference liquid water column on a single leaf or bare soil and $LAI$ the leaf area index. Exceeding liquid water is directly removed from the surface and infiltrated in the underlying soil. For paved surfaces, $m_\mathrm{liq,max}$ is set to $1\,\mathrm{mm}$. Exceeding liquid water is assumed to be drained off. Note that $m_\mathrm{liq}$ enters the calculation of $LE_\mathrm{liq}$ indirectly via $c_\mathrm{liq}$, which is given either as the ratio $m_\mathrm{liq}/m_\mathrm{liq,max}$ for vegetation (following the HTESSEL scheme) or $(m_\mathrm{liq}/m_\mathrm{liq,max})^{0.67}$
for pavement following Masson (2000) (based on Noilhan and Planton, 1989).



### 2.1.3 Calculation of resistances

The resistances are calculated separately for bare soil and vegetation following Jarvis (1976). The canopy resistance, $r_c$, is calculated as

$$r_c = \frac{r_{c,min}}{LAI} \, f_1(SW_\downarrow) \, f_2(\widetilde{m}) \, f_3(e_{def}),$$
(16)

with $r_{c,min}$ being a minimum canopy resistance. $f_1 - f_3$ are correction functions depending on $LAI$, the incoming shortwave radiation ($SW_\downarrow$) and the water-vapor pressure deficit ($e_{def} = e_{sat} - e$, with $e_{sat}$ and $e$ being the water-vapor pressure at saturation and the current water-vapor pressure, respectively). The layer-averaged volumetric soil moisture content ($\widetilde{m}$) is given by

$$\tilde{m} = \sum_{k=1}^{N} R_{fr,k} \, \max(m_{soil,k}, m_{wilt}),$$
(17)

where $N$ is the number of soil layers, $R_{fr,k}$ is the root fraction in layer $k$, $m_{soil,k}$ is the volumetric soil moisture content in layer $k$, and $m_{wilt}$ is the permanent wilting point.

The correction functions $f_1$ and $f_2$ read

$$\frac{1}{f_1(SW_\downarrow)} = \min\left(1, \frac{0.004 \, SW_\downarrow}{0.81(0.004 \, SW_\downarrow + 1)}\right),$$
(18)

which accounts for the reaction of plants to sunlight (opening/closing stomatas); the reaction of plants to water availability in 155   the soil is considered via

$$\frac{1}{f_2(\tilde{m})} = \begin{cases} 0 & \tilde{m} < m_{wilt} \\ \dfrac{\tilde{m} - m_{wilt}}{m_{fc} - m_{wilt}} & m_{wilt} \leq \tilde{m} \leq m_{fc} \\ 1 & \tilde{m} > m_{fc}, \end{cases}$$
(19)

with $m_{fc}$ being the soil moisture at field capacity. Furthermore, a correction for the water vapor pressure deficit is given by

$$\frac{1}{f_3(e_{def})} = \exp(g_D \, e_{def}),$$
(20)

where $g_D$ is a correction factor that is used for high vegetation.

The soil resistance ($r_{soil}$) is calculated as

$$r_{soil} = r_{soil,min} \cdot f_4(m_{soil,1}),$$
(21)

where $r_{soil,min}$ is the minimum soil resistance. The correction function $f_4$ is given by

$$f_4 = \max\left(\frac{m_{soil,1} - m_{min}}{m_{fc} - m_{min}}, 1\right),$$
(22)





with $m_{\text{min}}$ being a minimum soil moisture for the soil matrix based on the wilting point and the residual moisture $m_{\text{res}}$,
calculated as

$$m_{\text{min}} = c_{\text{veg}} \, m_{\text{wilt}} + (1 - c_{\text{veg}}) m_{\text{res}} \,. \tag{23}$$

Note that the total surface resistance ($r_{\text{s}}$, cf. Eq. (9)) is calculated as a diagnostic quantity from $LE$ after the energy balance
is solved.

## 2.2 Soil model

The soil model consists of prognostic equations for the soil temperature and the volumetric soil moisture which are solved for
multiple layers. The soil model only takes into account vertical transport within the soil and no ice phase is considered at the
moment. By default, the soil model consists of eight layers, with default layer depths of 0.01, 0.02, 0.04, 0.06, 0.14, 0.26, 0.54,
and 1.86 m, but the number of layers as well as their depths can be modified. The vertical heat and water transport is modeled
using the Fourier law of diffusion and Richards' equation, respectively. For vegetated surface elements, root fractions can be
assigned to each soil layer to account for the explicit water withdrawal of plants used for transpiration from the respective soil
layer. Viterbo and Beljaars (1995) and Balsamo et al. (2009) give more details

### 2.2.1 Soil heat transport

The Fourier law of diffusion reads

$$(\rho C)_{\text{soil}} \frac{\partial T_{\text{soil}}}{\partial t} = \frac{\partial}{\partial z} \left( \lambda_T \frac{\partial T_{\text{soil}}}{\partial z} \right) \,, \tag{24}$$

with $(\rho C)_{\text{soil}}$ and $\lambda_T$ being the volumetric heat capacity and the thermal conductivity of the soil layer in question, respectively.
$\lambda_T$ is calculated as

$$\lambda_T = Ke \, (\lambda_{T,\text{sat}} - \lambda_{T,\text{dry}}) + \lambda_{T,\text{dry}} \,, \tag{25}$$

with $\lambda_{T,\text{sat}}$, $\lambda_{T,\text{dry}}$, and $Ke$ being the thermal conductivity of saturated soil, of dry soil and the Kersten number, respectively.
$\lambda_{T,\text{sat}}$ is given by

$$\lambda_{T,\text{sat}} = \lambda_{T,\text{sm}}^{1-m_{\text{soil,sat}}} \, \lambda_{\text{m}} \,. \tag{26}$$

Here, $\lambda_{T,\text{sm}}$ is the thermal conductivity of the soil matrix and $\lambda_{\text{m}}$ is the heat conductivity of water. The Kersten number ($Ke$)
is calculated as

$$Ke = \log_{10} \left[ \max \left( 0.1, \frac{m_{\text{soil}}}{m_{\text{sat}}} \right) \right] + 1 \,. \tag{27}$$

At the bottom boundary a fixed deep soil temperature $T_{\text{deep}}$ is prescribed (Dirichlet conditions), which is a plausible assumption
for short term simulations covering only a few days.





**Table 1.** Look-up table for vegetation parameters of 18 predefined vegetation types in the style of the ECMWF-IFS classification, adapted for PALM. $\Lambda_{skin,s}$ and $\Lambda_{skin,u}$ are the total thermal conductivities between the skin layer and the surface for near-surface stable and unstable stratification, respectively. $\epsilon$ is the surface emissivity. All other symbols are used as defined in the main text.

| type | description | $r_{c,min}$ (m s⁻¹) | $LAI$ | $c_{veg}$ | $g_D$ (hPa⁻¹) | $z_0$ (m) | $z_{0,h}$ (m) | $\Lambda_{skin,s}$ (W m⁻² K⁻¹) | $\Lambda_{skin,u}$ (W m⁻² K⁻¹) | $C_0$ | albedo type | $\epsilon$ |
|---|---|---|---|---|---|---|---|---|---|---|---|---|
| 1 | bare soil | 0.0 | 0.00 | 0.00 | 0.00 | 0.005 | 0.5E-4 | 0.0 | 0.0 | 0.00 | 17 | 0.94 |
| 2 | crops, mixed farming | 180.0 | 3.00 | 1.00 | 0.00 | 0.10 | 0.001 | 10.0 | 10.0 | 0.00 | 2 | 0.95 |
| 3 | short grass | 110.0 | 2.00 | 1.00 | 0.00 | 0.03 | 0.3E-4 | 10.0 | 10.0 | 0.00 | 5 | 0.95 |
| 4 | evergreen needleleaf trees | 500.0 | 5.00 | 1.00 | 0.03 | 2.00 | 2.00 | 20.0 | 15.0 | 0.00 | 6 | 0.97 |
| 5 | deciduous needleleaf trees | 500.0 | 5.00 | 1.00 | 0.03 | 2.00 | 2.00 | 20.0 | 15.0 | 0.00 | 8 | 0.97 |
| 6 | evergreen broadleaf trees | 175.0 | 5.00 | 1.00 | 0.03 | 2.00 | 2.00 | 20.0 | 15.0 | 0.00 | 9 | 0.97 |
| 7 | deciduous broadleaf trees | 240.0 | 6.00 | 0.99 | 0.13 | 2.00 | 2.00 | 20.0 | 15.0 | 0.00 | 7 | 0.97 |
| 8 | tall grass | 100.0 | 2.00 | 0.70 | 0.00 | 0.47 | 0.47E-2 | 10.0 | 10.0 | 0.00 | 10 | 0.97 |
| 9 | desert | 250.0 | 0.05 | 0.00 | 0.00 | 0.013 | 0.013E-2 | 15.0 | 15.0 | 0.00 | 11 | 0.94 |
| 10 | tundra | 80.0 | 1.00 | 0.50 | 0.00 | 0.034 | 0.034E-2 | 10.0 | 10.0 | 0.00 | 13 | 0.97 |
| 11 | irrigated crops | 180.0 | 3.00 | 1.00 | 0.00 | 0.5 | 0.50E-2 | 10.0 | 10.0 | 0.00 | 2 | 0.97 |
| 12 | semidesert | 150.0 | 0.50 | 0.10 | 0.00 | 0.17 | 0.17E-2 | 10.0 | 10.0 | 0.00 | 11 | 0.97 |
| 13 | ice caps and glaciers | 0.0 | 0.00 | 0.00 | 0.00 | 1.3E-3 | 1.3E-4 | 58.0 | 58.0 | 0.00 | 14 | 0.97 |
| 14 | bogs and marshes | 240.0 | 4.00 | 0.60 | 0.00 | 0.83 | 0.83E-2 | 10.0 | 10.0 | 0.00 | 3 | 0.97 |
| 15 | evergreen shrubs | 225.0 | 3.00 | 0.50 | 0.00 | 0.10 | 0.10E-2 | 10.0 | 10.0 | 0.00 | 4 | 0.97 |
| 16 | deciduous shrubs | 225.0 | 1.50 | 0.50 | 0.00 | 0.25 | 0.25E-2 | 10.0 | 10.0 | 0.00 | 5 | 0.97 |
| 17 | mixed forest/woodland | 250.0 | 5.00 | 1.00 | 0.03 | 2.00 | 2.00 | 20.0 | 15.0 | 0.00 | 10 | 0.97 |
| 18 | interrupted forest | 175.0 | 2.50 | 1.00 | 0.03 | 1.10 | 1.10 | 20.0 | 15.0 | 0.00 | 7 | 0.97 |





**Table 2.** Look-up table for soil parameters.

| type | description | $\alpha$ | $l$ | $n$ | $\gamma_{\mathrm{sat}}$ $(\mathrm{m\,s^{-1}})$ | $m_{\mathrm{sat}}$ $(\mathrm{m^3\,s^{-3}})$ | $m_{\mathrm{fc}}$ $(\mathrm{m^3\,s^{-3}})$ | $m_{\mathrm{wilt}}$ $(\mathrm{m^3\,s^{-3}})$ | $m_{\mathrm{res}}$ $(\mathrm{m^3\,s^{-3}})$ |
|---|---|---|---|---|---|---|---|---|---|
| 1 | coarse | 3.83 | 1.150 | 1.38 | 6.94E-6 | 0.403 | 0.244 | 0.059 | 0.025 |
| 2 | medium | 3.14 | -2.342 | 1.28 | 1.16E-6 | 0.439 | 0.347 | 0.151 | 0.010 |
| 3 | medium-fine | 0.83 | -0.588 | 1.25 | 0.26E-6 | 0.430 | 0.383 | 0.133 | 0.010 |
| 4 | fine | 3.67 | -1.977 | 1.10 | 2.87E-6 | 0.520 | 0.448 | 0.279 | 0.010 |
| 5 | very fine | 2.65 | 2.500 | 1.10 | 1.74E-6 | 0.614 | 0.541 | 0.335 | 0.010 |
| 6 | organic | 1.30 | 0.400 | 1.20 | 1.20E-6 | 0.766 | 0.663 | 0.267 | 0.010 |

### 2.2.2 Soil moisture transport

The vertical transport of water within the soil matrix is calculated using Richards' equation, viz.

$$\frac{\partial m_{\mathrm{soil}}}{\partial t} = \frac{\partial}{\partial z}\left(\lambda_m \frac{\partial m_{\mathrm{soil}}}{\partial z} - \gamma\right) + S_m\,, \tag{28}$$

where $\lambda_m$, $\gamma$, and $S_m$ are the hydraulic diffusion coefficient, hydraulic conductivity, and a sink term due to root extraction, respectively. The hydraulic diffusion coefficient is calculated after Clapp and Hornberger (1978) as

$$\lambda_m = \frac{b\gamma_{\mathrm{sat}}(-\Psi_{\mathrm{sat}})}{m_{\mathrm{sat}}}\left(\frac{m_{\mathrm{soil}}}{m_{\mathrm{sat}}}\right)^{b+2}\,, \tag{29}$$

with $b = 6.04$ being a fixed parameter, $\gamma_{\mathrm{sat}}$ being the hydraulic conductivity at saturation, and $\Psi_{\mathrm{sat}} = -338$ m being the soil matrix potential at saturation. The hydraulic conductivity ($\gamma$) is calculated after van Genuchten (1980) (as in HTESSEL):

$$\gamma = \gamma_{\mathrm{sat}}\frac{\left[(1+(\alpha h)^n)^{1-1/n} - (\alpha h)^{n-1}\right]^2}{(1+(\alpha h)^n)^{(1-1/n)(l+2)}}\,. \tag{30}$$

Here, $\alpha$, $n$, and $l$ are van Genuchten coefficients that depend on the soil type (see Table 2). $h$ is the pressure head, which is calculated via rearrangement of

$$m_{\mathrm{soil}}(h) = m_{\mathrm{res}} + \frac{m_{\mathrm{sat}} - m_{\mathrm{res}}}{(1+(\alpha h)^n)^{1-1/n}}\,. \tag{31}$$

The root extraction of water from the respective soil layer $S_{m,k}$ is calculated as follows:

$$S_{m,k} = \frac{LE_{\mathrm{veg}}}{\rho_l\, l_{\mathrm{v}}}\frac{R_{\mathrm{fr},k}}{\Delta z_{\mathrm{soil},k}}\frac{m_{\mathrm{soil},k}}{m_{\mathrm{total}}}\,, \tag{32}$$

where $m_{\mathrm{total}}$ is the total water content of the soil,

$$m_{\mathrm{total}} = \sum_{k=1}^{N} R_{\mathrm{fr},k}\, m_{\mathrm{soil},k}\,, \tag{33}$$





**Table 3.** Look-up table for albedo parameters. (*1) land ice is treated differently than sea ice (*2) preliminary/dummy values

| albedo type | Description | broadband | longwave | shortwave | Notes |
|---|---|---|---|---|---|
| 1 | ocean | 0.06 | 0.06 | 0.06 | |
| 2 | mixed farming, tall grassland | 0.19 | 0.28 | 0.09 | |
| 3 | tall/medium grassland | 0.23 | 0.33 | 0.11 | |
| 4 | evergreen shrubland | 0.23 | 0.33 | 0.11 | |
| 5 | short grassland/meadow/shrubland | 0.25 | 0.34 | 0.14 | |
| 6 | evergreen needleleaf forest | 0.14 | 0.22 | 0.06 | |
| 7 | mixed deciduous forest | 0.17 | 0.27 | 0.06 | |
| 8 | deciduous forest | 0.19 | 0.31 | 0.06 | |
| 9 | tropical evergreen broadleaved forest | 0.14 | 0.22 | 0.06 | |
| 10 | medium/tall grassland/woodland | 0.18 | 0.28 | 0.06 | |
| 11 | desert, sandy | 0.43 | 0.51 | 0.35 | |
| 12 | desert, rocky | 0.32 | 0.40 | 0.24 | |
| 13 | tundra | 0.19 | 0.27 | 0.10 | |
| 14 | land ice | 0.77 | 0.65 | 0.90 | *1 |
| 15 | sea ice | 0.77 | 0.65 | 0.90 | |
| 16 | snow | 0.82 | 0.70 | 0.95 | |
| 17 | bare soil | 0.08 | 0.08 | 0.08 | |
| 18 | asphalt/concrete mix | 0.17 | 0.17 | 0.17 | *2 |
| 19 | asphalt (asphalt concrete) | 0.17 | 0.17 | 0.17 | *2 |
| 20 | concrete (Portland concrete) | 0.30 | 0.30 | 0.30 | *2 |
| 21 | sett | 0.17 | 0.17 | 0.17 | *2 |
| 22 | paving stones | 0.17 | 0.17 | 0.17 | *2 |
| 23 | cobblestone | 0.17 | 0.17 | 0.17 | *2 |
| 24 | metal | 0.17 | 0.17 | 0.17 | *2 |
| 25 | wood | 0.17 | 0.17 | 0.17 | *2 |
| 26 | gravel | 0.17 | 0.17 | 0.17 | *2 |
| 27 | fine gravel | 0.17 | 0.17 | 0.17 | *2 |
| 28 | pebblestone | 0.17 | 0.17 | 0.17 | *2 |
| 29 | woodchips | 0.17 | 0.17 | 0.17 | *2 |
| 30 | tartan (sports) | 0.17 | 0.17 | 0.17 | *2 |
| 31 | artificial turf (sports) | 0.17 | 0.17 | 0.17 | *2 |
| 32 | clay (sports) | 0.17 | 0.17 | 0.17 | *2 |
| 33 | building (dummy) | 0.17 | 0.17 | 0.17 | *2 |



with $R_{\mathrm{fr},k}$ being the root fraction in soil layer $k$. Only those layer are summed up which have a soil moisture above wilting point (i.e. plants are not able to withdraw water from layers with soil moisture below wilting point). The root distribution within the soil must chosen in such a way that

$$\sum_{k=1}^{N} R_{\mathrm{fr},k} = 1 \,. \tag{34}$$

There are two options available for the bottom boundary conditions for soil moisture. The bottom surface can be set to either bedrock, i.e. water can not be drained off and accumulated in the lowest soil layer (water content conservation), or it can be set to free drainage, i.e. an open bottom where soil water is continuously lost by drainage.

### 2.2.3 Treatment of pavements

Pavements are treated as a common soil (allowing varying number and depths of the pavement layers) but with physical properties of the pavement material. The pavement layer is impermeable to water and prohibits the vertical transport of soil moisture. Soil layers are placed below the pavement layers.

### 2.2.4 Treatment of water bodies

For water surfaces, PALM currently only allows for prescribing a bulk water temperature. The energy balance is then solved as for land surfaces, but without evapotranspiration from vegetation and bare soil (see above). A skin layer is adopted so that $C_0 = 0$ and $\Lambda = 1 \times 10^{11}$ in order to calculate the heat flux into the water body.

### 2.3 Numerical methods

In order to solve for the energy balance for the surface temperature ($T_0$), Eq. (1) is first linearized around $T_0$ at the current time step and then discretized in time using PALM's default Runge-Kutta third-order time stepping scheme. In this way, an iterative procedure to solve the energy balance is avoided and the prognostic equation then reads

$$T_0^{t+1} = \frac{A\Delta t + C_0 T_0^t}{C_0 + B\Delta t} \,, \tag{35}$$

where $t$ is the time index, $\Delta t$ is the current time step. $A$ and $B$ are coefficients given by

$$A = R_{\mathrm{n}} + 3\sigma T_0^4 + \left(\frac{\rho c_p}{r_{\mathrm{a}}}\right)\theta_{\mathrm{mo}} + \left(\frac{\rho l_{\mathrm{v}}}{r_{\mathrm{a}} + r_{\mathrm{s}}}\right)\left(q_{\mathrm{mo}} - q_{\mathrm{sat}} + \frac{dq_{\mathrm{sat}}}{dT}T_0\right) + \Lambda\,T_{\mathrm{soil},1} \tag{36}$$

and

$$B = \Lambda + \left(\frac{\rho l_{\mathrm{v}}}{r_{\mathrm{a}} + r_{\mathrm{s}}}\right)\frac{dq_{\mathrm{sat}}}{dT} + \left(\frac{\rho c_p}{r_{\mathrm{a}}\Pi}\right) + 4\sigma T_0^3 \,. \tag{37}$$

Here, $\sigma = 5.67037 \times 10^{-8}$ is the Stefan-Boltzmann constant. For vegetated surfaces, where $C_0$ is zero, Eq. (35) reduces to a diagnostic relationship viz.

$$T_0^{t+1} = \frac{A}{B} \,. \tag{38}$$



## 3   CESAR observations

For model evaluation we chose to simulate two consecutive clear-sky days observed on 5[th] and 6[th] of May, 2008, at the CESAR site at Cabauw. The period was chosen because the forcing from the surface was dominant and larger-scale advection played a minor role. We used direct measurements of temperature, humidity and wind and derived observations of sensible, latent and ground heat flux as well as net radiation. CESAR features a 213 m high measurement mast with instruments in 1.5, 10, 20, 40, 80, 140 and 200 m (Bosveld, 2020b). For temperature, humidity and wind, 10-minute averages are derived with a measurement

accuracy for temperature and humidity of $0.1\,\mathrm{K}$ and $3.5\,\%$, respectively (Meijer, 2000). Soil temperature is observed at a depth of 0, 2, 4, 6, 8, 12, 20, 30 and 50 cm. Additionally, as part of the IMPACT-EUCAARI campaign in May 2008 (Intensive Measurement Period at the Cabauw Tower within the European Integrated project on Aerosol Cloud Climate and Air Quality Interactions,  Kulmala et al., 2009), radiosondes were launched daily at 05:00, 10:00, and 16:00 UTC.

The surface soil heat flux is derived from soil temperature measurements by means of a Fourier extrapolation. Net radiation

is derived from the budget of the four radiation components (Eq. 2). Turbulent fluxes of sensible and latent heat are derived by means of the eddy-covariance (EC) method. In the process of calculating EC-fluxes from raw turbulence data it is unavoidable that low frequency contributions to the flux are getting lost. For the current Cabauw data, 10 minute intervals are used with simple subtraction of the mean from the raw turbulent timeseries; afterwards a low frequency correction is applied based on the spectra by Kaimal et al. (1972), taking into account wind speed and stability (Bosveld, 2020a). The method of low-frequency

loss correction assumes that all turbulence characteristics follow surface-layer scaling. However, this is not always true, as for example horizontal advection by organized turbulent structures (Eder et al., 2015; De Roo and Mauder, 2018) may add further low-frequency contributions which is not accounted for in the surface-layer scaling. For further information on the instrumentation of the Cabauw site we refer to Bosveld (2020b).

Cabauw is located in the western part of the Netherlands and surrounded by mostly meadows with ditches passing through

as well as villages, orchards and lines of trees. The CESAR tower itself is installed over an area of short grass that is kept at a height of about 8 cm. The immediate surroundings of the measuring tower are free of significant heterogeneities for a few hundreds of meters. During the simulation period from 5[th] to 6[th] of May, 2008 the prevailing wind direction is from south-east and the 10 m average wind speed ranges from 2 to $6\,\mathrm{m\,s^{-1}}$. The convective boundary layer reaches a height of around $2\,\mathrm{km}$. The groundwater level is in 1.3 m depth. The profiles of temperature $\theta$, humidity $q$ and horizontal wind $v_\mathrm{h}$ retrieved from

radiosounding are shown in Fig. 2 together with tower measurements from all available levels (see details in Sect. 5). The soil temperature $T_\mathrm{soil}$ and soil moisture $m_\mathrm{soil}$ at 2008-05-05 05:00 UTC is depicted in Fig. 1 (see Sect. 4).

## 4   Simulation setup

To evaluate a land-surface parameterization scheme the relevant vegetation and soil information is required. With regards to this, the CESAR site is well described in literature (e.g. Beljaars and Bosveld, 1997). The reference simulation (here-

after referred to as case REF) was set up as a best guess with the Cabauw land surface parameters according to Beljaars and Bosveld (1997), Ek and Holtslag (2004) and Maronga and Bosveld (2017). The vegetation type of the surface is 'short





grass' with some modifications. The roughness length for momentum is set to $0.15\,\mathrm{m}$, which is representative for a few kilometers of upstream terrain from the Cabauw tower and the roughness length for temperature is set to $2.35 \times 10^{-5}\,\mathrm{m}$ (Ek and Holtslag, 2004). The leaf area index is $1.7\,\mathrm{m^2\,m^{-2}}$, the minimum canopy resistance $110\,\mathrm{s\,m^{-1}}$ and we chose a

heat conductivity between skin layer and soil of $4\,\mathrm{W\,m^2\,K^{-1}}$. The heat capacity of the skin layer is set to $0\,\mathrm{J\,m^{-2}\,K^{-1}}$, the surface emissivity $\epsilon = 0.97$ and the surface is covered to $100\,\%$ with vegetation. The soil layers are defined at depth of $0.005, 0.02, 0.04, 0.065, 0.1, 0.15, 0.24, 0.45, 0.675, 1.125$ and $2.25\,\mathrm{m}$. The soil parameters for field capacity and wilting point are $0.491$ and $0.314$, respectively, which according to the ECMWF-IFS classification would best be described as very fine soil. The residual moisture is set to $0.01\,\mathrm{m^3\,m^{-3}}$ and the minimum soil resistance to $50\,\mathrm{s\,m^{-1}}$. The deep soil temperature is fixed

at $283.19\,\mathrm{K}$ which is a valid assumption, because the lower two soil levels are not reached by diurnal temperature variations (lower part of Fig. 1 does not change over time). The van Genuchten coefficients, the hydraulic conductivity at saturation and the porosity vary with soil depth. In the uppermost $24\,\mathrm{cm}$ the parameters are set to match medium-fine soil (type 3, in Table 2), the layer between $24$-$60\,\mathrm{cm}$ is the same as the uppermost layer except for the porosity which had to be increased due to observed large values of soil moisture, and between $60$-$225\,\mathrm{cm}$ the parameters are set to organic soil according to Table 2. In all

simulations, the land surface and soil parameters are homogeneous over the model domain. This means that buildings of the small town of Lopik, west of the CESAR tower, are neglected, as well as the small ditches which cross the observation site.

The root fraction and initial soil profiles of temperature and moisture are shown in Fig. 1. In $3\,\mathrm{cm}$ depth begins a layer of relatively high root density $R_{\mathrm{dens}}$ down to $18\,\mathrm{cm}$ followed by a layer of relatively low root density down to $60\,\mathrm{cm}$ depth. According to Jager et al. (1976), no roots are found near the surface ($< 3\,\mathrm{cm}$) and in the deep soil layers ($> 60\,\mathrm{cm}$). The

initial soil temperature and moisture profiles are taken from measurement data at 2008-05-05 05:00 UTC (shown in Fig. 1). A summary of the land surface scheme configuration and its used values are listed in Table 4.

Case REF is driven by external forcing of incoming short- and longwave radiation taken from the Cabauw measurements. In this way, the effects of high altitude aerosols, moisture and clouds are included. Accordingly, the degree of freedom is reduced and we can focus on parameters of the LSM rather than additional uncertainties of a radiation model. Nonetheless,

we performed sensitivity tests using the Rapid Radiation Transfer Model for Global Models (RRTMG, Clough et al., 2005) as well as a clear-sky radiation parameterization, which are described in detail in Maronga et al. (2020). The longwave outgoing radiation of the surface is calculated from the skin-layer temperature using Stefan-Boltzmann law. The long- and shortwave albedos of diffusive radiation are set to $0.34$ and $0.14$, respectively, to fit the dominating grassland. Albedos of direct radiation are calculated according to Briegleb (1992) considering a weak solar zenith-angle dependence as such that their direct values

equal the diffusive ones at $60\,^{\circ}$.

The model domain for case REF is $(x \times y \times z)$ $2000\,\mathrm{m} \times 2000\,\mathrm{m} \times 4317\,\mathrm{m}$ with a horizontal and vertical grid spacing of $50\,\mathrm{m}$ and $10\,\mathrm{m}$, respectively. A grid sensitivity study was carried out to justify this choice (see Sect. 5). Starting at $2000\,\mathrm{m}$, i.e. above the boundary-layer top, a vertical grid stretching is applied with a stretching factor of $1.08$ and a maximum vertical grid spacing of $100\,\mathrm{m}$. Initial profiles of temperature, humidity and horizontal wind of case REF are taken from radiosounding data

and are shown in Fig. 2. The horizontal wind equals the initial $u$ component of the wind vector, i.e. at the beginning of the





**Table 4.** Overview of the land surface scheme configuration for case REF.

| Parameter | Value | Description |
|---|---|---|
| **Skin layer parameters** | | |
| $C_0$ | $0\,\mathrm{J\,m^{-2}\,K^{-1}}$ | Heat capacity of the skin layer |
| $c_{\mathrm{veg}}$ | $100\,\%$ | Vegetation coverage of the surface |
| $LAI$ | $1.7$ | Leaf area index |
| $r_{\mathrm{c,min}}$ | $110\,\mathrm{s\,m^{-1}}$ | Minimum canopy resistance |
| $z_0$ | $0.15\,\mathrm{m}$ | Roughness length for momentum |
| $z_{0,\mathrm{h}}$ | $2.35 \times 10^{-5}\,\mathrm{m}$ | Roughness length for temperature |
| $\Lambda_{\mathrm{skin}}$ | $4.0$ | Heat conductivity between skin layer and soil |
| $\epsilon$ | $0.97$ | Surface emissivity |
| **Soil parameters** | | |
| $m_{\mathrm{res}}$ | $0.010\,\mathrm{m^3\,s^{-3}}$ | Residual volumetric soil moisture |
| $r_{\mathrm{soil,min}}$ | $50\,\mathrm{s\,m^{-1}}$ | Minimum soil resistance |
| $T_{\mathrm{deep}}$ | $283.19\,\mathrm{K}$ | Deep soil temperature |
| $m_{\mathrm{fc}}$ | $0.491\,\mathrm{m^3\,s^{-3}}$ | Volumetric soil moisture at field capacity |
| $m_{\mathrm{wilt}}$ | $0.314\,\mathrm{m^3\,s^{-3}}$ | Volumetric soil moisture at permanent wilting point |
| **Height dependent soil parameters (0-24cm, 24-60 cm, 60-225 cm)** | | |
| $\alpha$ | $0.83, 0.83, 1.30$ | van Genuchten coefficient |
| $l$ | $-0.588, -0.588, 0.400$ | van Genuchten coefficient |
| $n$ | $1.25, 1.25, 1.20$ | van Genuchten coefficient |
| $\gamma_{\mathrm{sat}}$ | $0.26, 0.26, 1.2 \times 10^{-6}\,\mathrm{m\,s^{-1}}$ | Hydraulic conductivity of the soil at saturation |
| $m_{\mathrm{sat}}$ | $0.430, 0.766, 0.766\,\mathrm{m^3\,s^{-3}}$ | Volumetric soil moisture at saturation (porosity) |
| **Initial soil profiles** | | |
| $T_{\mathrm{soil},k}$ | $283.96, 284.00, 284.62, 284.59, 284.70, 284.77,$ $284.55, 283.50, 283.19, 283.19, 283.19\,\mathrm{K}$ | Soil temperature at depth level $k(k \in 1, 11)$ |
| $m_{\mathrm{soil},k}$ | $0.324, 0.324, 0.324, 0.332, 0.352, 0.380, 0.477,$ $0.605, 0.670, 0.721, 0.721\,\mathrm{m^3\,m^{-3}}$ | Soil moisture at depth level $k(k \in 1, 11)$ |
| $R_{\mathrm{fr},k}$ | $0, 0, 0.2, 0.2, 0.2, 0.2, 0.1, 0.1, 0, 0, 0$ | Root fraction at depth level $k(k \in 1, 11)$ |

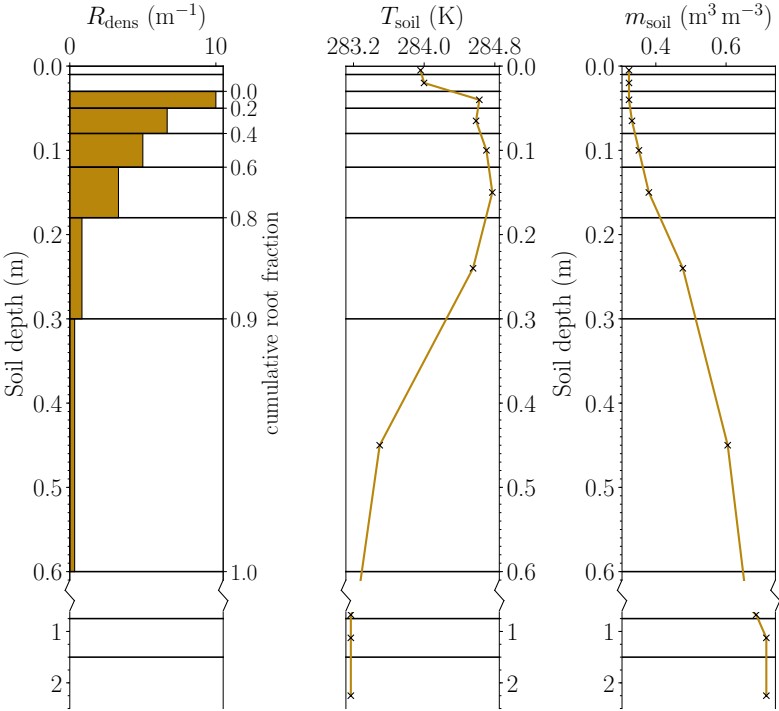

**Figure 1.** Vertical soil layer setup of root density ($R_{\mathrm{dens}}$, left) as well as initial profiles of temperature (middle) and moisture (right). Note the broken vertical axis with a changed linear increment in the deeper layers. The root fraction of each soil layer (cf. $R_{\mathrm{fr}}$ in Table 4) is the difference of the cumulative root fraction (Eq. (34), shown on the vertical axis) between two layers.

simulation there is no wind turning with height due to Coriolis force. The geostrophic wind at the domain top is set to $7\,\mathrm{m\,s^{-1}}$ and $0\,\mathrm{m\,s^{-1}}$ for the $u$ and $v$ component, respectively. The lateral boundary conditions are cyclic.

All sensitivity simulations are based on the setup of the reference simulation and only differ by the respective parameter to be analyzed. An overview of the sensitivity simulations and their well-defined change compared to case REF is given in
Table 5.

## 5 Results

### 5.1 Boundary-layer profiles

At first, we will look at the vertical profiles of the radiosonde and the tower measurements. Figure 2 shows vertical profiles of $\theta$, $q$ and $v_{\mathrm{h}}$ indicating the evolution of the boundary layer during the considered period of time in Cabauw at the times
of the radiosonde ascents. Both nights show a stable nocturnal boundary layer before sunrise (at $05{:}00\,\mathrm{UTC}$). A nighttime low-level jet is observed in the horizontal wind. Profiles of potential temperature and water vapor mixing ratio show a well-mixed convective boundary layer at $10{:}00\,\mathrm{UTC}$ and $16{:}00\,\mathrm{UTC}$ of either day. Over the course of the two days, the depth of the



**Table 5.** Overview of the case study and their changes relative to case REF

| Case | changes to case REF |
|---|---|
| REF | - |
| ALBE_24 | shortwave albedo (at $60°$) of $0.24$ |
| ALBE_44 | shortwave albedo (at $60°$) of $0.44$ |
| ADV_tq | advection of $T$ and $q$ in all heights according to mean change in radiosounding data between $2.5\,\mathrm{km}$ and $4\,\mathrm{km}$ |
| CAP_2e4 | $C_0 = 2 \times 10^4\,\mathrm{J\,m^{-2}\,K^{-1}}$ |
| CLEARSKY | clear-sky radiation model |
| COND_2 | $\Lambda_{\mathrm{skin}} = 2\,\mathrm{W\,m^2\,K^{-1}}$ |
| COND_6 | $\Lambda_{\mathrm{skin}} = 6\,\mathrm{W\,m^2\,K^{-1}}$ |
| Dz_2 | $\Delta z = 2\,\mathrm{m}$ and $\Delta x = \Delta y = 5\,\mathrm{m}$ |
| EMIS_95 | $\epsilon = 0.95$ |
| EMIS_100 | $\epsilon = 1.00$ |
| HUMID_dry | initialization with $q_{\mathrm{v,k}} = 0$ |
| HUMID_sat | initialization with $q_{\mathrm{v,k}} = q_{\mathrm{v,sat}}$ |
| LAI_05 | $LAI = 0.5\,\mathrm{m^2\,m^{-2}}$ |
| LAI_3 | $LAI = 3\,\mathrm{m^2\,m^{-2}}$ |
| ROUGH_01 | $z_0 = 0.01\,\mathrm{m}$ and $z_{0,\mathrm{h}} = 1.57 \times 10^{-6}\,\mathrm{m}$ |
| ROUGH_001 | $z_0 = 0.1\,\mathrm{m}$ and $z_{0,\mathrm{h}} = 1.57 \times 10^{-7}\,\mathrm{m}$ |
| RRTMG | Rapid Radiation Transfer Model for Global Models (RRTMG) |
| SOIL_2 | $\alpha_{\mathrm{VG}}, l_{\mathrm{VG}}, n_{\mathrm{VG}}, \gamma_{\mathrm{sat}}$ as in soil type 2 (in the uppermost $60\,\mathrm{cm}$) |
| SOIL_4 | $\alpha_{\mathrm{VG}}, l_{\mathrm{VG}}, n_{\mathrm{VG}}, \gamma_{\mathrm{sat}}$ as in soil type 4 (in the uppermost $60\,\mathrm{cm}$) |
| TEMP_9 | initialization with $T_0 = 280.15$ (ca. $9°\mathrm{C}$) |
| TEMP_11 | initialization with $T_0 = 286.15$ (ca. $11°\mathrm{C}$) |

convective boundary layer increases. The profiles of $v_{\mathrm{h}}$ show a mismatch between the radiosounding measurements and the tower data. This is explained by different analysis timescales. One the one hand, the radiosonde records instantaneous values

with a sampling frequency of $0.1\,\mathrm{Hz}$, which results in five recording heights below $300\,\mathrm{m}$. On the other hand, data from the cup anemometers at the mast is temporally averaged over $10\,\mathrm{min}$, based on a $3\,\mathrm{s}$ running mean calculated with an update frequency of $4\,\mathrm{Hz}$ (Bosveld, 2020b). Thus, the mean tower profiles can be over- or underestimated by the instantaneous values of the radiosonde. Note that we initialize the simulations with data from the radiosonde ascent, because it reaches high enough, but the comparison later is made against tower data due to its higher temporal resolution.

Figure 2 also shows the vertical profiles of the reference simulation, which are temporally averaged over $15\,\mathrm{min}$ and horizontally averaged over the whole domain. A comparison of observed profiles of $\theta$ and $q$ with those of the LES show that on the first day, the simulated boundary layer is too shallow at $10{:}00\,\mathrm{UTC}$. One hypothesis to explain this is that the turbulence development during model spin-up, i.e. in the first morning, is slower than in reality. At this time, the horizontal wind speed



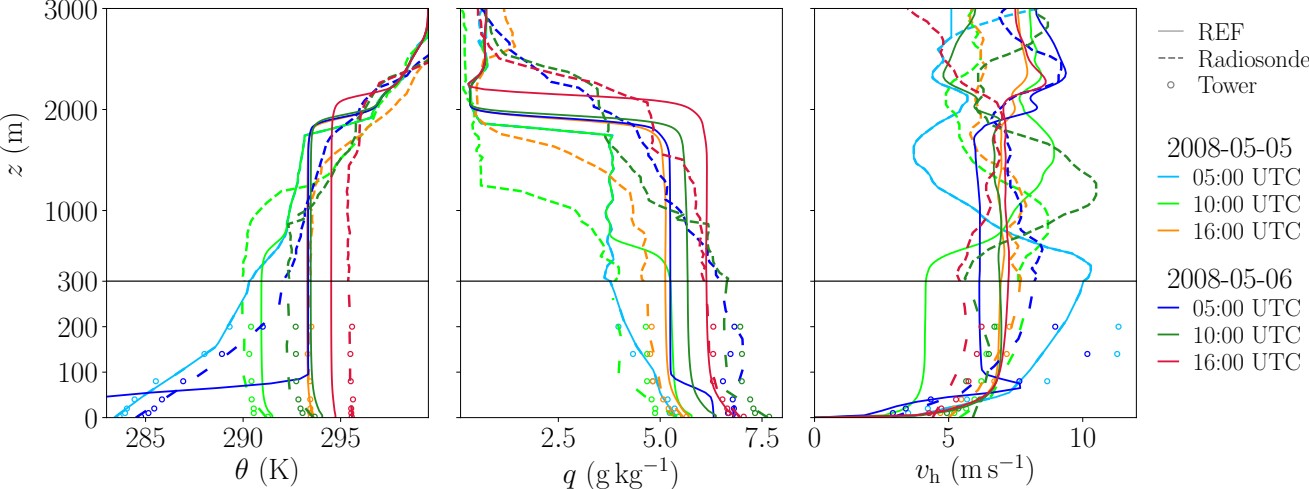

**Figure 2.** Vertical profiles of $\theta$, $q$ and $v_\mathrm{h}$ measured by radiosonde (dashed lines) and tower (point markers) at the CESAR site as well as simulated profiles of case REF (continuous lines) during the two days period. Note that the lower $300\,\mathrm{m}$ are shown with a higher vertical resolution than the layers above to better visualize individual tower measurement (black horizontal line indicates the break). In the left panel, the orange line is partially hidden behind the blue one.

shows no good agreement between model and observations. At 16:00 UTC of the first day, the simulation slightly overestimates

the boundary layer depth despite a fairly good agreement of model and observations regarding mixed-layer temperature and humidity values. The wind profiles, however, agree well. At night (2008-05-06 05:00 UTC), the near-surface temperature is significantly lower (out of range at ca. $281\,\mathrm{K}$) than measured. Another difference is that the nocturnal boundary layer is too shallow in the simulation. At the same time, the near-surface humidity shows small differences between model and reality. The simulated horizontal wind speed also depicts a low-level jet, but compared with observations it occurs closer to the surface, in

accordance with the simulated nocturnal boundary-layer depth. Even though the LES cannot reproduce the nocturnal situation very well, the mixed-layer quickly develops in the next morning (2008-05-06 10:00 UTC) which is in agreement with findings of van Stratum and Stevens (2015). At 10:00 UTC of the second day the simulation indicates a warmer and deeper boundary layer compared to the observation, which could be caused by advection processes in reality modifying the residual layer and thus the boundary-layer evolution during the morning transition. The wind profiles agree fairly well. The temperature pro-

file at 2008-05-06 16:00 UTC shows that the simulation underestimates the mixed-layer temperature but instead has a higher boundary-layer top, suggesting that the energy input into the boundary layer is similar, but distributed over a deeper layer. The humidity profiles agree well in the lowest $1000\,\mathrm{m}$, but deviate above in accordance with the differences in boundary-layer depth. The horizontal wind is slightly overestimated. In general, the simulated profiles are much more constant with height in the well-mixed layer, because they depict domain averages, as opposed to the local measurements, so that a direct comparison

is inherently improper. Above the boundary layer, in the free atmosphere, synoptic-scale processes dominate in reality. Since we did not consider these in our simulations, profiles may deviate. Given the fact that the surface forcing in this particular





case was the dominant forcing for the development of the boundary layer, this deficiency should not compromise the present evaluation study.

## 5.2 Evaluation of energy-balance components

Figure 3 shows the time series of surface net radiation which follows a diurnal cycle typical for clear-sky conditions. At noon, the reference simulation underestimates $R_n$, whereas it overestimates $R_n$ at night, i.e. it is less negative by approximately $30\,\mathrm{W\,m^{-2}}$. The nocturnal differences are due to much lower surface temperatures of $280\,\mathrm{K}$ in case REF vs. $285\,\mathrm{K}$ in the observations (not shown). Since we prescribe the incoming radiation in case REF, the long-wave outgoing radiation flux is underestimated. At around 12:00 UTC of the second day, clouds influence the net radiation which is indicated by fluctuations

in the curve. Since case REF is driven with $SW_\downarrow$ taken from CESAR data, this cloud effect on the surface net radiation is included and can also be seen in the other surface fluxes (cf. Fig. 4, 5, 6). Figure 3 also depicts the effect of different land-surface properties and simulation setups on the surface radiation. Please note that we will only highlight the cases which lead to the largest differences compared to case REF for the respective variable. In Fig. 3, these are mostly the changes to the albedo and the radiation models as well as cases LAI_05 and HUMID_sat. The most obvious differences are seen in case of a changed

albedo. An increase of the albedo (case ALBE_44) leads to a decrease in $R_n$ and vice versa (case ALBE_24) with maximum deviation to case REF of about $\pm100\,\mathrm{W\,m^{-2}}$ at noon, indicating that mismatches in the estimated albedo cause significant errors on the surface net radiation during daytime. However, also the spread of the non-highlighted simulations (gray) is about $50\,\mathrm{W\,m^{-2}}$ at noon, meaning that also variations in the surface parameters (e.g. emissivity, roughness length) affect the surface net radiation significantly. Using the RRTMG, radiative fluxes are calculated for each horizontal grid box of the LES based

on time and geographical location as well as air pressure, and local profiles of temperature and humidity (Clough et al., 2005) instead of being taken from the measurements. One key feature of the RRTMG, which is neither included in observations nor in the clear-sky model, is the direct cooling of air due to longwave radiative flux divergence, particularly during nighttime. This can result in cooling rates in the order of $0.1$ to $0.3\,\mathrm{K\,h^{-1}}$ (see, e.g. Maronga and Bosveld, 2017, their Fig. 1). In case RRTMG, the simulated $R_n$ approaches that of the observations during day and night, though it slightly underestimates the

surface net radiation at noon by about $30\,\mathrm{W\,m^{-2}}$. However, we have to note that we use default input of the RRTMG with standard profiles of water vapor, other trace gases, and aerosol concentration above model top, which do not necessarily reflect the real conditions at Cabauw during the simulated time period and may impact the simulated surface net radiation, too. Using the clear-sky model, the simulated $R_n$ becomes even better during day, but also marginally overestimates net radiation during night by $10$ to $20\,\mathrm{W\,m^{-2}}$. Case LAI_05 has a smaller $R_n$ compared to case REF, because a smaller $LAI$ significantly decreases

$LE$ and increases $H$, which leads to higher surface temperatures during day resulting in higher $LW_\uparrow$. In case of a saturated humidity mixing-ratio (case HUMID_sat) $LE$, $H$ and $G$ are significantly altered in a way that higher surface temperature is simulated during day and night. This again leads to higher $LW_\uparrow$ and hence lower $R_n$. Except for ALBE_24 the simulations tend to underestimate the surface net radiation during daytime while the simulated surface net radiation tend to overestimate the observed one during night.



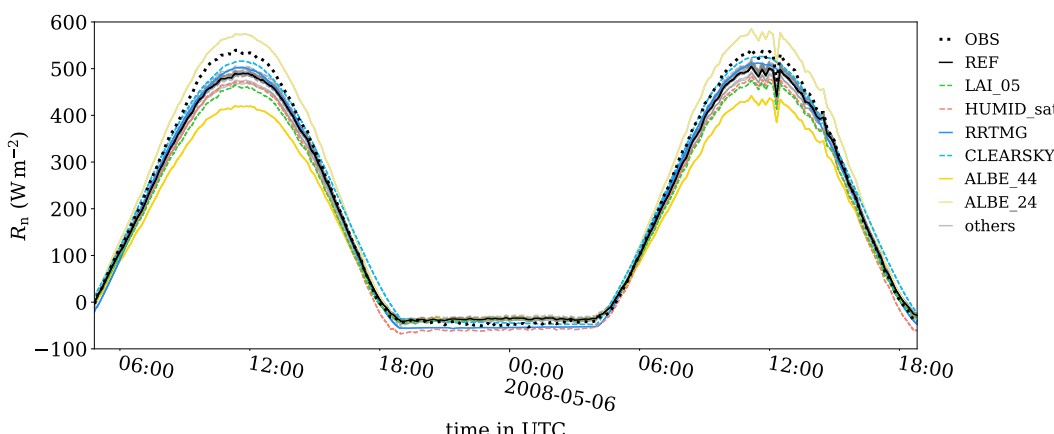

**Figure 3.** Timeseries of surface net radiation $R_\mathrm{n}$ as measured at Cabauw (OBS) and domain-averaged $R_\mathrm{n}$ for all simulation cases listed in Table 5. Only case REF and some relevant cases are highlighted, all others are shown in gray.

Figure 4 shows that the observed surface sensible heat flux reaches at maximum approx. $100\,\mathrm{W\,m^{-2}}$ and at minimum $-50\,\mathrm{W\,m^{-2}}$. Compared to observations (OBS), case REF significantly overestimates $H$, especially at noon by approximately $40\,\mathrm{W\,m^{-2}}$. Moreover, the simulated $H$ differs from observations in the morning and afternoon hours, where it is positive earlier and later, respectively. At night, case REF shows a constantly increasing $H$ from $-25$ to $-15\,\mathrm{W\,m^{-2}}$, whereas the observation show a secondary minimum of ca. $-50\,\mathrm{W\,m^{-2}}$ between $20{:}00\,\mathrm{UTC}$ and $23{:}00\,\mathrm{UTC}$. Here, we note that the depicted sensible

heat flux from the simulations are domain-averaged values. Considered locally in the simulation, we can detect similar temporal fluctuations in $H$, though with a smaller amplitude than those observed at the CESAR site. Like case REF, all sensitivity simulations tend to overestimate the observed heat flux, while the spread among the considered cases is significant. At noon, where the spread among all simulations is largest, it reaches values up to $120\,\mathrm{W\,m^{-2}}$. ALBE_44, LAI_3, and HUMID_dry best meet the observations. With a higher albedo (ALBE_44) the available energy at the surface becomes lower (cf. Fig. 3) leading

to smaller fluxes of $H$ and $LE$. Besides changing the albedo, case LAI_3 and LAI_05 show the largest impact on $H$, as with lower and higher $LAI$ the available energy is preferentially partitioned into $H$ and $LE$, respectively. Similarly, the available energy is also preferentially partitioned into $H$ and $LE$ for humid and dry air, indicated by HUMID_sat and HUMID_dry, respectively.

Figure 5 shows timeseries of the latent heat flux. The observations range from $0$ during night to about $300\,\mathrm{W\,m^{-2}}$ at noon.

Case REF matches the observation reasonably well during day- and nighttime, even though it overestimates $LE$ during the second day. Having a lower $LE$ than case REF, ALBE_44 best meets the observed $LE$ of the second day. Besides this, case LAI_05 significantly underestimates $LE$ by preferentially partitioning the available energy into $H$ (Fig. 4). Moreover, case HUMID_sat stands out because it shows a negative $LE$ during night, which is explained by dew formation in the model. Compared with the observations, where negative $LE$ is not observed, this suggests that dew formation has not been observed

in Cabauw. The peak spread of all non-highlighted sensitivity studies is around $50\,\mathrm{W\,m^{-2}}$ and up to $170\,\mathrm{W\,m^{-2}}$ of the high-





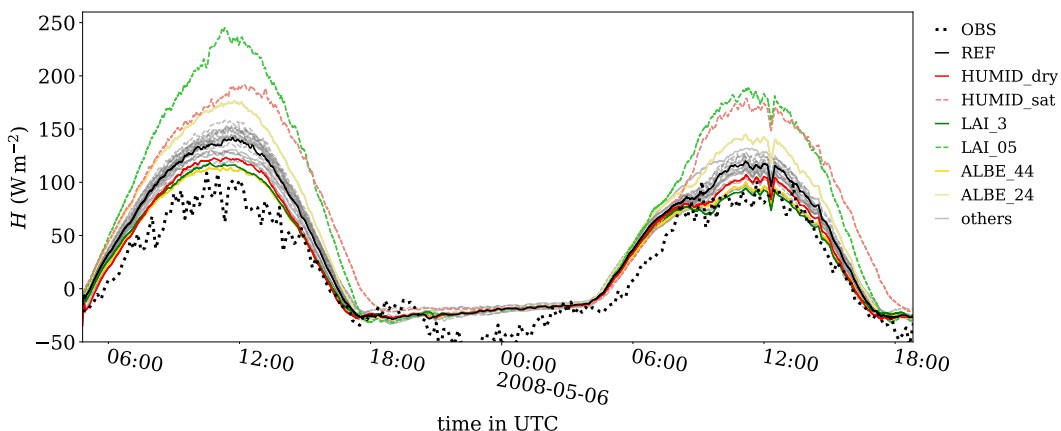

**Figure 4.** Timeseries of surface sensible heat flux $H$ as measured at Cabauw (OBS) and domain averaged $H$ for all simulation cases listed in Table 5. Only case REF and some relevant cases are highlighted, all others are shown in gray.

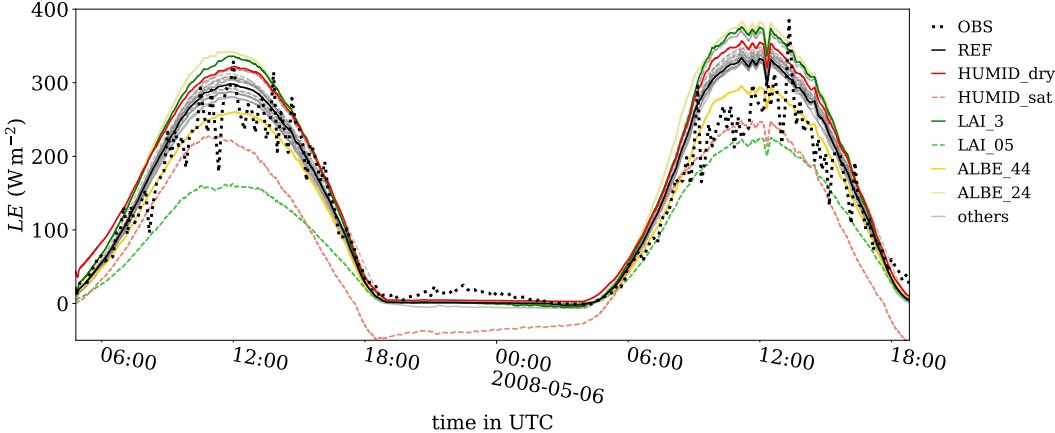

**Figure 5.** Timeseries of latent heat flux $LE$ as measured at Cabauw (OBS) and domain-averaged $LE$ for all simulated cases listed in Table 5. Only case REF and some relevant cases are highlighted, all others are shown in gray.

lighted. From Fig. 4 and Fig. 5 we can calculate the daytime Bowen ratio $\beta_0 = \frac{H}{LE}$ (not shown). We find that the difference between the Bowen ratio of the simulations and that observed at Cabauw ($\beta_{0,\mathrm{sim}} - \beta_{0,\mathrm{obs}}$) ranges from $-0.1$ to $+0.2$, except for cases HUMID_sat and LAI_05, which show significantly lower Bowen ratios compared to observations. As opposed to other models, which were shown to systematically overestimate the Bowen ratio measured in Cabauw during summer (Schulz et al., 1998; Sheppard and Wild, 2002), we cannot identify a systematic bias for Bowen ratio of PALM's LSM by means of the analyzed period.

Figure 6 shows the timeseries of the ground heat flux, which reaches $60\,\mathrm{W\,m}^{-2}$ during daytime in the observations. With respect to the amplitude of $G$, case REF shows good agreement with the observation. However, the shape of the timeseries is





discernibly different. At daytime, OBS shows a more sinusoidal shape, whereas the simulations show a more humped-shaped
diurnal variation. We attribute this to the method to derive the observed ground heat flux, which involves the average soil heat
flux in $10\,\mathrm{cm}$ and $5\,\mathrm{cm}$ depth and the soil temperature difference between $0$ and $2\,\mathrm{cm}$ (Bosveld, 2020b). In the model, on the
other hand, the ground heat flux is parameterized according to Eq. (6) and thus only the temperature gradient between the
surface and $0.5\,\mathrm{cm}$ are taken into account. Hence, the simulated $G$ resembles the diurnal cycle of the surface temperature,
while the observed $G$ correlates with the soil temperature. Observations and case REF agree well at noon with values $G$ around
$55\,\mathrm{W\,m^{-2}}$, whereas in the afternoon at 15:00 UTC case REF is almost $10\,\mathrm{W\,m^{-2}}$ higher than the observations. It strikes that all
simulations have a fairly small spread at that point in time. During the day, changing the heat conductivity between skin layer
and uppermost soil layer has the largest impact on the ground heat flux, with smaller and larger $G$ observed in case COND_2
and COND_6, respectively. This can be attributed to the linear relationship between $G$ and $\Lambda$ in Eq. (6). Also, the simulation
cases with a different $LAI$ show a relatively large deviation from case REF at noon, with increased (decreased) $G$ for LAI_05
(LAI_3). For example, with a smaller leaf area and thus less transpiration the available energy is preferentially partitioned
into $G$ and $H$ (see Fig. 4) rather than into $LE$. At noon, the spread among all simulations is about $40\,\mathrm{W\,m^{-2}}$, whereas the
non-highlighted cases show a maximum deviation from REF of no more than $\pm10\,\mathrm{W\,m^{-2}}$. In the night, cases HUMID_sat
and RRTMG stand out. In both cases the controlling variable is the (near-)surface temperature (see Fig. 9). In case RRTMG,
the surface cools down while the soil temperature does not change in the same amount (see Fig. 8), which leads to a strong
ground heat flux directed towards the surface. In case HUMID_sat, the surface stays relatively warm, therefore it results in a
small negative $G$. Except for cases HUMID_sat and COND_2, the model tends to simulate a stronger upward directed ground
heat flux at night. Compared to the observations, case COND_2 overestimates $G$ during night whereas it underestimates $G$ at
noon, which points to a stability dependence of the conductivity $\lambda$. Even though this is technically realized in the code, the
standard values for short grass do not differ between stable and unstable conditions due to a lack of knowledge of the correct
relationship between $\lambda$ and stability. As cases COND_2 and COND_6 do not have a significant impact on any of the other
variables, we can infer that uncertainties in modeling $G$ is relatively small. Nonetheless, please note that we only analyzed a
short period of two days. For longer simulation times covering multiple days, e.g. for heat-wave scenarios where heat storage
within the soil becomes important, this might become relevant again. Furthermore, it shall be mentioned that the ground heat
flux is small compared to the surface net radiation as well as the surface sensible and latent heat fluxes.

Similar to many other sites, also eddy-covariance measurements at the Cabauw site suffer from the well-known problem
of energy-balance non-closure (de Roode et al., 2010). To date, the leading hypothesis is that low-frequency contributions
are inherently not captured by the eddy-covariance method, leading to the situation that the sum of the surface heat fluxes
underestimates the available energy by $10$ to $30\,\%$ (Foken et al., 2011). Figure 7 shows the timeseries of the residual as
well as the individual differences between the simulated (case REF) and observed energy balance components. Note that the
shown residual is effectively the residual of the observations, because the energy balance in the model is zero at all times
(according to Eq. (1) with $C_0 = 0\,\mathrm{J\,m^{-2}\,K^{-1}}$ ). The residual exhibits a diurnal cycle with only small positive values of about
$15\,\mathrm{W\,m^{-2}}$ during nighttime, while during daytime the residual indicates that up to $150\,\mathrm{W\,m^{-2}}$ are missing in order to close the
surface energy balance. The differences between the simulated and observed $H$ and $LE$ correlate negatively with the residual,



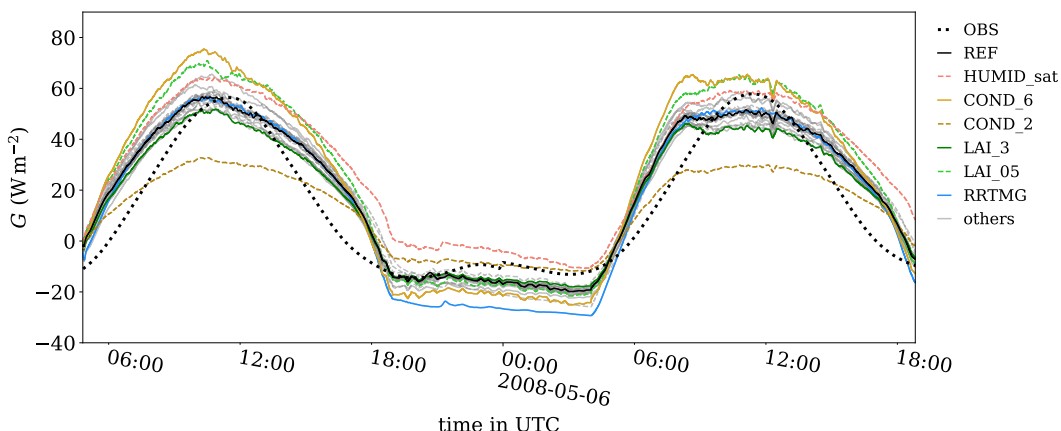

**Figure 6.** Timeseries of ground heat flux $G$ as measured at Cabauw (OBS) and domain-averaged $G$ for all simulated cases listed in Table 5. Only case REF and some relevant cases are highlighted, all others are shown in gray.

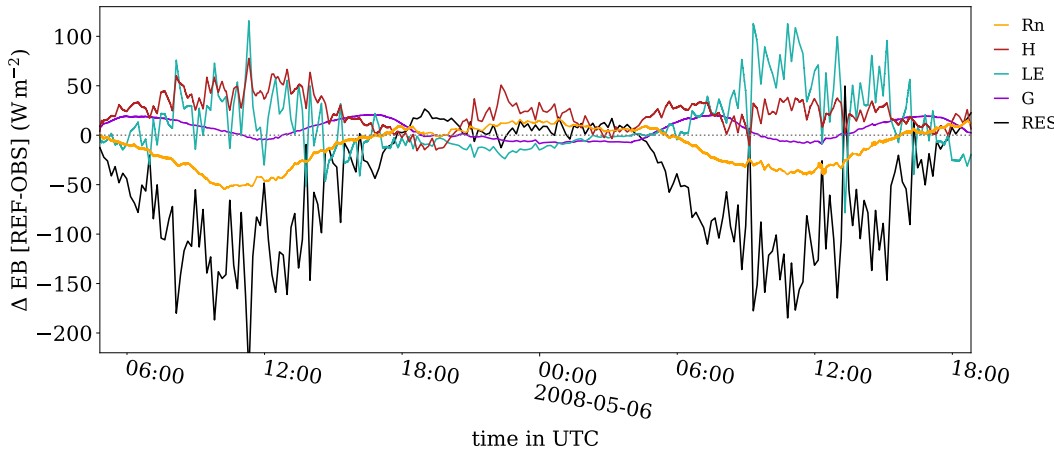

**Figure 7.** Timeseries of the individual differences between case REF and observations for the energy balance (EB) components $R_\mathrm{n}$, $H$, $LE$, and $G$, as well as the residual $RES = R_\mathrm{n} - H - LE - G$ from the Cabauw observations.

which indicates that the simulated $H$ and $LE$ are overestimated compared to the observations. In reverse, however, this may also suggest that the observed $H$ and $LE$ are underestimated, even though we explicitly note that we cannot know from the simulations how the missing energy is partitioned onto the individual energy-balance components in reality as both, observation and simulation may contain a bias. Taking this into account, we summarize that the four components of the energy balance are represented reasonably well by the LES-LSM interface. Also, the daytime Bowen ratio agrees fairly wells with that observed at Cabauw. By contrast, climate models have often shown to overestimate the Bowen ratio in summertime (Schulz et al., 1998; Sheppard and Wild, 2002).



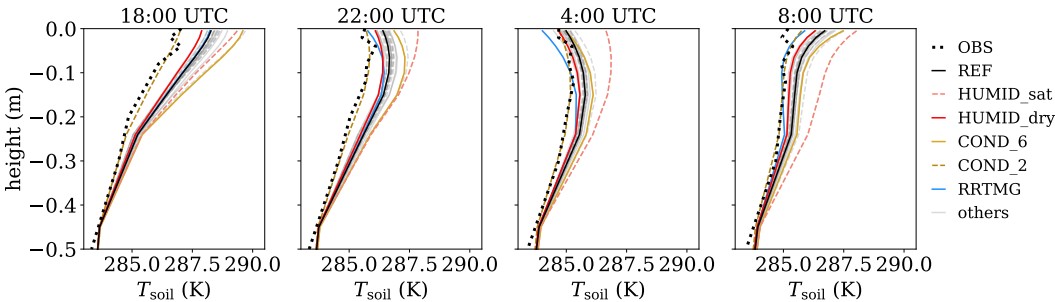

**Figure 8.** Vertical profiles of soil temperature as measured at Cabauw (OBS) and domain-averaged soil temperature for all simulated cases listed in Table 5. Only case REF and some relevant cases are highlighted, all others are shown in gray.

Figure 8 shows mean profiles of soil temperature $T_{\mathrm{soil}}$ averaged over the whole domain and over 15 min. Please note that there are doubts about the quality of the CESAR soil temperature observations, because of problems during calibration (Bosveld, 2020b) and the speculation that the sensors had sunken deeper into the soil over time (Bosveld, 2020a). Hence, observations cannot be used as a reliable reference to evaluate the model. Nevertheless, we will discuss the variation among
the different simulations. In the late afternoon of the first day (18:00 UTC), case REF shows a continuous decrease with depth from 288 K close to the surface to 283.5 K in −50 cm. During night, the soil begins to cool down starting at the surface, where it reaches 285 K at 04:00 UTC. The maximum soil temperature is now found in a depth of −15 cm. As soon as the net radiation becomes positive (at 08:00 UTC), the soil temperature close to the surface follows such that the maximum is again found close to the surface. The parameters with the largest spread - up to 4 K during day and 2 K during night - are, like for $G$, the
conductivity and atmospheric humidity. The non-highlighted cases have a spread of about 1 K indication that e.g. changing radiation or vegetation parameters have only minor effect on the soil temperature. A higher conductivity (COND_6) leads to higher soil temperatures, especially in the late afternoon (18:00 UTC), because the heat of the surface net radiation is more easily conducted into the soil (cf. Fig. 6); correspondingly COND_2 has smaller $T_{\mathrm{soil}}$. During night, the lower atmosphere in case HUMID_sat does not cool down as much as case REF (cf. Fig. 9), therefore the soil remains relatively warm, too.

**5.3 Evaluation of atmospheric quantities**

Next, we will evaluate the quantities characterizing the lower atmosphere, that is temperature, humidity, and wind speed. First, we will discuss the nighttime situation followed by the daytime situation. Finally, results from the sensitivity study will be discussed. Figure 9 shows the vertical profiles of potential temperature during night. At 18:00 UTC, the lowest levels start to cool, while the upper levels still indicate a vertically well-mixed state; observation and case REF agree well. At 22:00 UTC,
the stable boundary layer of case REF has reached a height of about 75 m and has a mean potential temperature gradient of 0.1 K m$^{-1}$. Tower measurements, however, show that it was less stable (0.02 K m$^{-1}$) and the stable layer is at least 200 m deep (not shown), above the mast we have no data to compare it with. The radiosounding profiles at 05:00 UTC in Fig. 2 show that there is no residual layer, but the stable layer extends up to the capping inversion at about $z = 1800$ m. The next morning, at



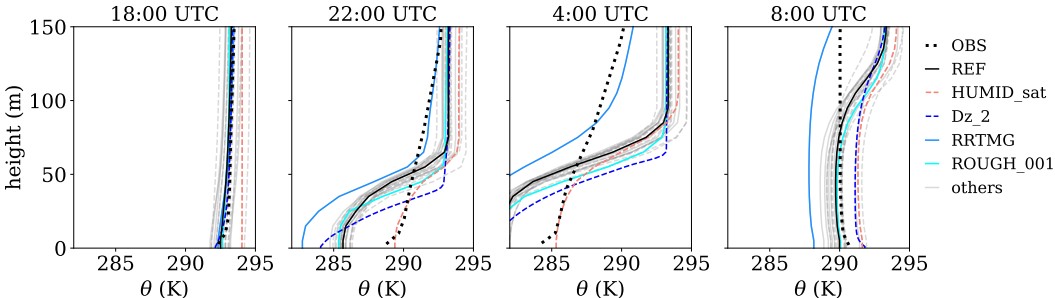

**Figure 9.** Vertical profiles of mean potential temperature as measured at Cabauw (OBS) and for all cases listed in Table 5. Only case REF and some relevant cases are highlighted, all others are shown in gray.

08:00 UTC, observations shows an already vertically well-mixed lower boundary layer, while in the simulations a stable layer
at about $100\,\mathrm{m}$ is still present and gets eroded due to surface heating and mixing processes. This is consistent with Fig. 11 which indicates a faster temperature increase at $z = 40\,\mathrm{m}$ compared to the observation.

One reason for the misrepresentation of the nocturnal stable layer could be too coarse grid spacing. However, compared to case REF, case Dz_2 with smaller grid spacing is even more stable during the night hours, having a lower boundary layer depth and cooler near-surface temperature in agreement with van Stratum and Stevens (2015). This suggests that the coarse resolution
might not explain the misrepresentation of the stable boundary layer, nonetheless with only one grid sensitivity simulation, a non-linear relationship cannot be fully ruled out. Another parameter affecting the diffusivity is the subgrid-scale scheme of the LES. Yet, the subgrid-scale scheme of Dai et al. (2020), which is more diffusive in the middle of the stable boundary layer but less diffusive towards the surface, does not change the stability of the simulation (not shown). Alternatively, too low diffusivity could be due to low wind speed, however the simulated wind speed in Fig. 13 agrees well with observations. Note, as $LW_\downarrow$
is prescribed in case REF we can exclude all factors affecting the longwave-incoming radiation, e.g. humidity or clouds, as a possible reason for the misrepresentation of the nighttime stable layer. At 02:00 UTC, case RRMTG, where cooling of the air column by vertical divergence of radiative fluxes is directly considered, reveals an even cooler and more stable (compared to REF) nighttime boundary layer. Too much cooling indicates too small humidity in the atmosphere, which is in agreement to Fig. 12, where humidity is slightly smaller in case RRMTG compared to the observations between 00:00 UTC and 06:00 UTC.
Furthermore, compared to the observation, the simulations indicate a less negative $H$ (see Fig. 4), though we note that the simulated $H$ shows a domain-averaged value while the observed value is taken from a point measurement with more temporal fluctuation. Nevertheless, according to the less negative $H$, the near-surface layer should be less stable in the simulations, which, however, is in contrast to the profiles shown in Fig. 11, which would suggest that the simulated $H$ should show larger negative values.
Figure 10 shows vertical profiles of horizontal wind speed $v_\mathrm{h}$. Again, we find that case REF and all sensitivity studies simulate a shallower nocturnal boundary layer than observed at the CESAR tower (22:00 UTC and 04:00 UTC). A low-level jet develops in reality as well as in the simulations. From the observational data it is deduced that the maximum wind speed



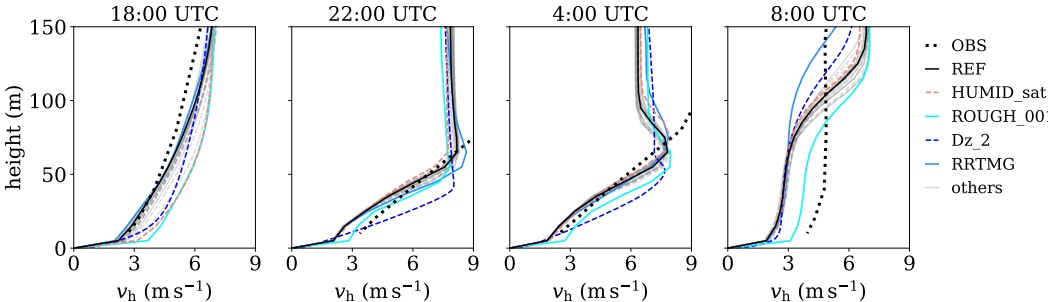

**Figure 10.** Vertical profiles of mean horizontal wind speed as measured at Cabauw (OBS) and for all cases listed in Table 5. Only case REF and some relevant cases are highlighted, all others are shown in gray.

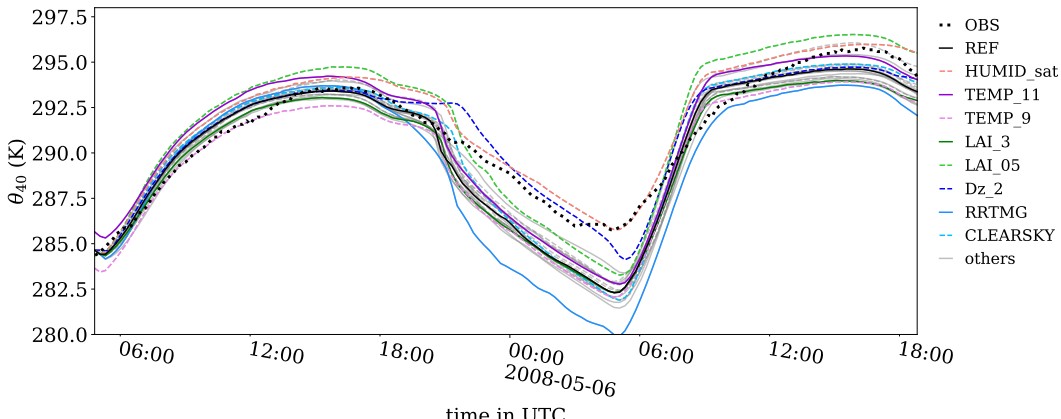

**Figure 11.** Timeseries of mean potential temperature at $40\,\mathrm{m}$ as measured at Cabauw (OBS) and for all cases listed in Table 5. Only case REF and some relevant cases are highlighted, all others are shown in gray.

occurs in a height above the tower. According to the radiosounding measurements at 2008-05-06 05:00 UTC (Fig. 2), the maximum wind speed occurs in approximately $500\,\mathrm{m}$. In the LES (case REF), however, the low-level jet has its maximum
between $60\,\mathrm{m}$ and $70\,\mathrm{m}$ height (cf. Fig. 2, 10). In the morning, when convective mixing sets in, the wind profiles become more well-mixed. At 08:00 UTC, we already find a mixed-layer in reality. By contrast, the stable layer has not been fully eroded in the LES, therefore the wind speed in the residual layer (above ca. $100\,\mathrm{m}$) is still higher than that of the lower $100\,\mathrm{m}$. Even though case RRTMG shows a large influence on the whole temperature profile (Fig. 9), its wind profile only deviates from case REF above the low-level jet and during the morning transition where the stable layer is already further eroded. Likewise, case
HUMID_sat does not deviate much in Fig. 10. Conversely, case ROUGH_001 deviates significantly in the wind profiles but does not show significant changes in the temperature profiles (Fig. 9). Only case Dz_2 shows a strong interconnection between temperature and wind profiles.



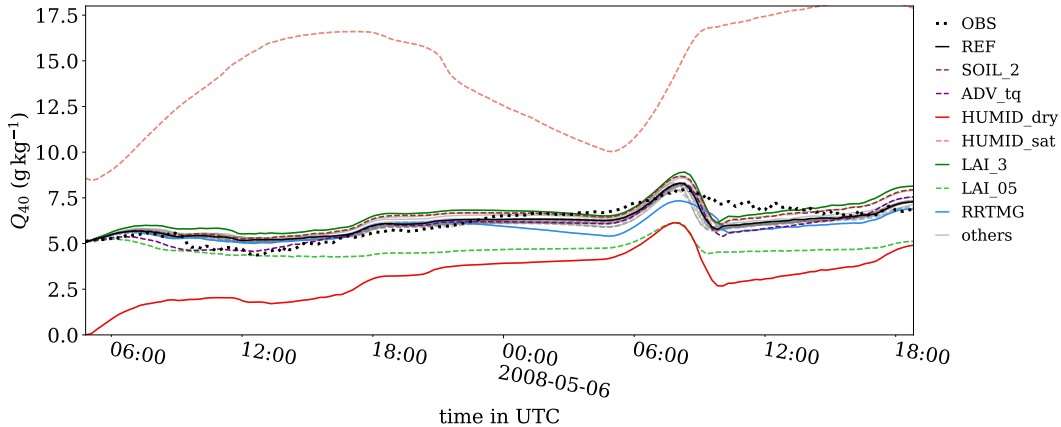

**Figure 12.** Timeseries of mean total water mixing ratio at $40\,\mathrm{m}$ as measured at Cabauw (OBS) and for all cases listed in Table 5. Only case REF and some relevant cases are highlighted, all others are shown in gray.

The CESAR tower samples temperature, humidity and horizontal wind speed in 10, 40, and $80\,\mathrm{m}$ height (wind speed not available at $1.5\,\mathrm{m}$). We chose to compare timeseries of the $40\,\mathrm{m}$ level, because on the one hand, $10\,\mathrm{m}$ lies between the first and second grid level and on the other hand, $80\,\mathrm{m}$ might already be above the nocturnal boundary layer (see discussion of Fig. 9). Figure 11 shows the diurnal cycle of temperature at $z = 40\,\mathrm{m}$ which, at first sight, agrees fairly well between observations and case REF. The temperature amplitude is around $10\,\mathrm{K}$ and between 16:00 and 17:00 UTC peak temperatures of $295\,\mathrm{K}$ and $296\,\mathrm{K}$ are reached on the first and second day, respectively. Significant differences are, however, found during night as well as during the second day: During night, case REF becomes more stable compared to the observations (Fig. 9), hence, the temperature in $40\,\mathrm{m}$ height is lower compared to the observations, as discussed earlier. The following morning, the boundary layer is heated up quickly between roughly $06{:}00\,\mathrm{UTC}$ and $08{:}00\,\mathrm{UTC}$ in the simulation, whereas in reality temperature increases slower. Again, the reason is that case REF develops a much shallower stable boundary layer (with a residual layer above, cf. Fig. 2). Once the stable layer is eroded up to $40\,\mathrm{m}$ the temperature can increase rapidly at that level as it is heated from the surface as well as from above, where warmer air from the residual layer is mixed into the shallow layer. In reality, where the stable layer is much deeper, the air in $40\,\mathrm{m}$ height is first only heated from the surface and not by entrainment. The sensitivity study shows that during night case HUMID_sat is in agreement with the observations at that height (cf. Fig. 9). This is because the humidity reaches saturation and condenses on the vegetation as dew, and heat from condensation is released to the air by different partitioning of the surface fluxes. However, dew formation was not observed at the CESAR site as shown by timeseries of $LE$ (Fig. 5), thus humidity is not suitable to explain the misrepresentation of near-surface temperature. In Fig. 11, also case Dz_2 seems to agree with observations. However, as shown in Fig. 9, this is only true for the depicted height of $40\,\mathrm{m}$, e.g. below $20\,\mathrm{m}$, the nocturnal air temperature is lower compared to case REF. Even though case Dz_2 has a much higher resolution, we cannot be certain that turbulence in the nocturnal boundary layer is sufficiently resolved and hence the resolution is still a possible explanation for the misrepresentation of the stable boundary layer in the LES. Another parameter to point out in Fig. 11 is case



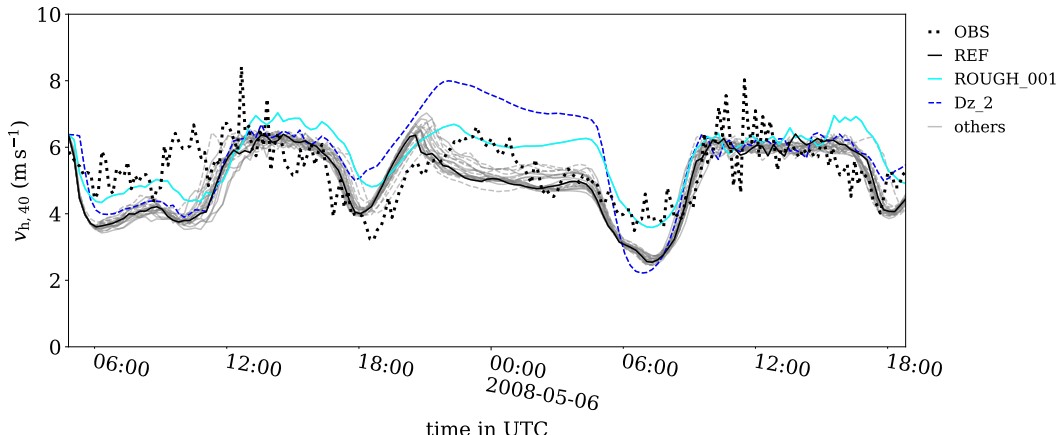

**Figure 13.** Timeseries of mean horizontal wind speed at $40\,\mathrm{m}$ as measured at Cabauw (OBS) and for all cases listed in Table 5. Only case REF and some relevant cases are highlighted, all others are shown in gray.

RRTMG, which shows significant differences to case REF during night, i.e. lower temperature and higher boundary layer. The
following day, the mixed-layer temperature of case RRTMG is persistently smaller compared to case REF due to the additional
cooling of the air volume at night but similar boundary-layer heating. This misrepresentation propagates into the next day.

Figure 12 shows that the observed humidity in $40\,\mathrm{m}$ has only small diurnal variations. The reference simulation mostly
agrees with the observations. The small decrease in the measurement data around $09{:}00\,\mathrm{UTC}$ of the first day can be reproduced
by case ADV_tq, which includes advection tendencies of $\theta$ and $q$ according to the mean change well above the boundary
layer. However, during the morning transition of the second day (around $08{:}00\,\mathrm{UTC}$) the humidity in the simulations first rises
and then drops while near-surface air is mixed with dry air from above. By contrast, this is not observed in the measurement
data, most likely because no strong stable layer had developed during nighttime. The majority of the simulated cases do not
show a large spread (about $0.5\,\mathrm{g\,kg^{-1}}$), except for cases HUMID_dry and HUMID_sat. Case HUMID_dry, which is initialized
with zero humidity, but becomes humid through the latent heat flux from the surface, follows a similar curve as case REF
but with persistently lower values. On the other hand, case HUMID_sat, initialized with saturation moisture, shows a diurnal
cycle similar to that of potential temperature. Apart from the extreme cases of initialized humidity, the choice of the radiation
model and $LAI$ reveal the most significant deviation from case REF. If the diurnal cycles of temperature and humidity are
placed in context with those of sensible and latent heat fluxes, it is found that while the fluxes are consistently overestimated
during day, corresponding higher values of $\theta$ or $q$ are not observed. This can be explained with differences between observation
and simulation in the boundary-layer depth. Instead, the observed differences in the atmosphere occur during night or in the
morning and are attributed to a misrepresentation of the nocturnal boundary layer.

The timeseries of observed horizontal wind speed, depicted in Fig. 13, show significant amount of fluctuation during day and
some during night. Since the LES data shown is horizontally averaged over the entire horizontal domain, the fluctuations are
less prominent in the simulations. Nevertheless, the mean daytime and nighttime magnitude agrees reasonably well with the





observations, including the decreases and consecutive increases in the evening (ca. 18:00 UTC) and morning (ca. 06:00 UTC). Relevant deviations from case REF are found if the roughness length is reduced, i.e. case ROUGH_001. As expected, this leads to higher wind speed close to the surface. Reducing the grid size (case DZ_2) has the same effect mostly shown at night.

## 6   Summary

In this paper we gave a description of the land-surface model embedded into the PALM model system which is applied to model

the surface energy balance at vegetated or paved land surfaces as well as at water surfaces. We evaluated the land-surface model implementation against in-situ observations of the energy-balance components as well as near-surface wind-, temperature-, and humidity-profiles taken at Cabauw over a quasi homogeneous flat grass site (Monna and Bosveld, 2013) for two consecutive diurnal cycles. A sensitivity study showed the relative importance of the choice of land-surface input parameters and thereby gives valuable reference for the user.

The diurnal cycles of surface latent and sensible heat flux are well represented. Even though the model seems to overestimate the fluxes, the differences can be explained by the non-closure of observed energy balance, i.e. missing energy in $H$ and $LE$ observed at the CESAR site. During daytime the simulated Bowen ratio agrees reasonably well with the observed one, whereas climate models overestimate the summer Bowen ratios observed in Cabauw (e.g. Schulz et al., 1998; Sheppard and Wild, 2002). The diurnal cycle of the modeled ground heat flux agrees with the observations, though the modeled flux overestimates

the observed flux during the morning and evening hours. During nighttime, the modeled ground heat flux shows slightly larger negative values compared to the observation. Due to its relatively small contribution compared to the surface net radiation or the surface latent and sensible heat fluxes, these mismatches do not affect boundary-layer development too much, if only one or two days are simulated. However, for longer simulation periods heat storage in the soil may become an important factor, e.g. when heat waves built-up over days (Miralles et al., 2014). During daytime the model tends to underestimate the surface net

radiation, while during nighttime the surface net radiation shows slightly less negative values compared to the observations, indicating underestimated longwave radiative cooling. The near-surface temperature matches well with the observed one at the first simulated day. During nighttime, however, the near-surface temperature is underestimated in the model, the nocturnal boundary layer is too stable and too shallow compared to the observations. The diurnal cycle of the near-surface wind is well represented, though the low-level jet in the model occurs much closer to the surface.

In addition to the evaluation against observations, we carried out a comprehensive sensitivity study. Land-surface parameters and the initial state of the atmosphere were varied within a typical range for the respective quantity. The net radiation is significantly influence by the albedo, the radiation model as well as many of the land surface parameters. The ground heat flux, though not as important as the other energy-balance components, is mostly influenced by the soil conductivity. The distribution of the available energy into surface sensible and latent heat fluxes depends mostly on the leaf area index as well as the initial

atmospheric humidity. Within the investigated range of $LAI$ values for short grass, differences of up to 50 % are possible. Moderately less relative deviation is found for the range of completely dry to saturated initial atmospheric humidity. Potential temperature is influenced by a change in $LAI$ of $\pm 2\,\mathrm{m}^2\,\mathrm{m}^{-2}$ as much as it is influenced by an initial temperature deviation





in the whole atmosphere of $\pm 1\,\mathrm{K}$. While most of the sensitivity studies show the most significant difference during day, the choice of initial humidity, grid size, roughness length and radiation model plays an important role at night. Overall, we could not identify a single parameter as being the most sensitive in all quantities at the same time. In fact, different parameters become relevant if different quantities are analyzed.

In order to evaluate and possibly improve land-surface schemes also for different types of surfaces, e.g. pavements or different vegetation coverage, reliable measurements of the energy-balance terms at different surfaces types are required. However, eddy-covariance measurements often suffer from the well-known problem of energy-balance non-closure, where the sum of surface sensible, latent, and ground heat flux underestimates the surface net radiation by about 10 to 30 % (Foken et al., 2011). Besides measurement uncertainties and footprint biases, one leading hypothesis is that low-frequency contributions from organized turbulent structures or surface heterogeneity-induced circulations are inherently not captured by the eddy-covariance method (e.g. Finnigan et al., 2003; Foken, 2008; Eder et al., 2015). As the modeled energy balance is closed by definition, it is thus difficult to draw final conclusions that may point directions for further improvements of the land-surface parameterizations as both, the model as well as the observation may contain a bias.

As the description of the land-surface model embedded in PALM only reflects its current state, a short outlook into future development is given below. Until now, the LSM implementation does not incorporate a tile approach (as the embedded building-surface model (Resler et al., 2017) does), such that a land-surface grid cell in PALM is either classified as water or as pavement-/vegetation-covered. Particularly for coarser grids, however, patchy landscapes such as e.g. with small ditches like in Cabauw, might be filtered, meaning that the relative contributions of surface types and thus the area-averaged energy-balance terms become a function of the horizontal grid size. In order to avoid this, one of our next steps will be to implement a tile approach into the land-surface model. Furthermore, the current LSM implementation does not include heat storage within water bodies. However, especially for multi-day simulations, e.g., for heat-wave scenarios, heat storage in water bodies might become important to accurately represent the cool-air production in urban environments. Hence, we plan to improve the representation of water surfaces by implementing a lake parameterization (e.g. Mironov et al., 2010). Further lines of future development will be the implementation of a snow parameterization as well as a parameterization to consider phase transitions to also consider frozen soil. In this study we only considered a homogeneously flat surface. Nonetheless, heterogeneous surfaces as well as step-like orography is implemented in the LSM. In addition, we plan to implement an immersed boundary method (Mason and Sykes, 1978) where elevation changes can also be represented by slanted surfaces.

*Code and data availability.* The PALM model system is freely available from http://palm-model.org and distributed under the GNU General Public License v3 (http://www.gnu.org/copyleft/gpl.html). The code version used and all input files are permanently available under https://doi.org/10.25835/0005529 (Gehrke et al., 2020).



*Author contributions.* Implementation of the LSM: BM; Conceptualization of the study: BM; investigation, formal analysis, and visualization: KFG; methodology, software, writing original draft, review and editing of writing: all authors.

*Competing interests.* The authors declare that they have no conflict of interest.

*Acknowledgements.* BM would like to thank Chiel van Heerwaarden (Wageningen University, Netherlands), Anton Beljaars (ECMWF, Reading, UK) and Fred Bosveld (KNMI, de Bilt, Netherlands) for many fruitful discussions during the land surface model implementation phase. BM and MS were funded by the German Federal Ministry of Education and Research (BMBF) under grant 01LP1601 within the framework of Research for Sustainable Development (FONA; www.fona.de). All simulations with PALM have been performed at the
supercomputers of HLRN, which is gratefully acknowledged.





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
