# Peer review of "Modeling of land-surface interactions in the PALM model system 6.0: Land surface model description, first evaluation, and sensitivity to model parameters"

_Geoscientific Model Development, 2020_

## Referee Comment (RC1) · Anonymous Referee #1 · 9 Nov 2020

Within this study, the authors introduce the land-surface model (LSM) implemented in PALM model, and evaluate the performance using two-day in-situ observations. They conduct a series of sensitivity experiments to explore the impacts of model parameters on simulating the boundary-layer profiles, the surface energy balance, and near-surface meteorological variables.

Despite the detailed description about the LSM and useful information for PALM users, the results are very preliminary. As the manuscript reads now, the authors touched some subjects only briefly without really adding any scientific merit. It is more like a

graduate's project essay than a scientific paper.

**Specific comments:**

The manuscript needs significant restructuring in order to establish a better focus within this paper. I suggest the authors seriously consider the sensitivity experiment design first. Of course every parameter in a model would have more or less impacts on the simulations. To fit this study, no need to take the radiation scheme (RRTMG & CLEARSKY), large-scale forcing (ADV_tq), or resolution (Dz_2) into account. Other case like EMIS_95 and EMIS_100, even not being mentioned in the manuscript. These unnecessary results make the paper more difficult to follow and do not have much scientific merit being covered so briefly. Removing the relevant content would be better, in my opinion.

Second, the results section seems poorly phrased. I see a little conjecture and repetition in Section 5. For example in L430-445, this portion could be removed (at least be shortened), as it does not provide much "facts" to convince readers. If I correctly understand, the point is the observed H and LE might be underestimated due to the limitation of eddy-covariance method, which partially explains the overestimated H and LE by model. Fig. 7 can be removed as well because we've already got those information from Figs. 3-6. The black line of RES term just indicates the measurements are of bad quality. Plus, a repeated statement about Bowen ratio in the end; authors have mentioned that in L396-400. For L472-489, after going through this paragraph, I still have no idea why the model is not able to reproduce the nocturnal boundary layer, even feel a big unsolved issue existed in the LSM or the atmospheric model. Authors should not do like give a hypothesis, reject it, and then say we in actual don't have much confidence in the rejection. This discussion won't help raise one's interest in the model.

The third point is to be correct. Like L355, I assume ALBE_24 is the sensitivity experiment featuring a decreased albedo in comparison to the REF. But following L232, the

shortwave albedo is set to 0.14 in REF. Please double check the setup of your experiments. L398: I doubt the discussion "cases HUMID_sat and LAI_05 show significantly lower Bowen ratios compared to observations". From Figs 4&5, I see larger H and lower LE which means a larger Bowen ratio (H/LE) than observations. In L509, it is the low temperature leading to stable boundary layer, not "stable layer, hence the low temperature". Likewise later in L513, the convective boundary layer started developing because of the surface heating in the morning than "the stable layer is eroded and temperature can rapidly increase".

Lastly, seriously improve the English writing.

**Technical comments**

L1: PALM is an acronym?

L2: "For this" -> "To this end"

L4: Add "with observations" after "agree well"

L8 & L47: "By this" -> "In this way"

L235: What is CESAR?

L263-264: Rephrase the sentence to "The CESAR site is well equipped with the vegetation and soil information which provides a good opportunity to evaluate the land-surface parameterization proposed in the present study. "

L267-271: Change to "The land surface scheme configuration is given in Table 4" and then add the information you don't have in Table 4.

L314: "One the one hand" -> "On one hand"

Fig.2: Crowded figure. May be plotted as Fig. 9, one time in one panel.

L326, L334 & L337: Add "with observations" after "agree well"

L377: "Moreover, the simulated H ..., respectively" -> "The model overestimates H

during the daytime while the maximum bias is at noon."

L394: "Compared with ... in Cabauw" -> "However, the observed LH is non-negative which suggests the dew formation does not occur in Cabauw."

L550: "at vegetated ... water surfaces" -> "at vegetated surface"
* * *

---

## Referee Comment (RC2) · Anonymous Referee #2 · 3 Dec 2020

Summary:
* * *
This paper describes the land-surface model in the PALM model system. The subject matter is approprite for the GMD journal and the text is well written. The details about the model are provided in appropriate amount of detail, equations provided are also appropriate. As part of the paper, there is a "first evaluation" which I think could/should be improved upon, as detailed in my comments below. Even though this is a "first

evaluation" I think it can be done with more rigor and there are, in my opinion, a few technical flaws. While I think the description of the model is ready to be published, the evaluation of the model (and comparison with observations) needs improvement before that portion of the paper should be published.

General Comments:
* * *
1. Most land-surface models vs observation comparisons I am aware of use the model run in "single-point" mode with the observed tower data driving the model. However, the model output from the PALM LSM has been averaged over the spatial domain. This spatial seems to confound the comparison (e.g., l.337-340). Is there a reason the spatial averaging of the model data is necessary?

2. The 2-day period seems too short to do a thorough evaluation of the model. There are many decades of Cabauw data, but only a short 2-day period is used. Even for a "first evaluation" this seems like a weakness of the paper. According to the authors, this particular period was chosen was because (l.236) "...the forcing from the surface was dominant and larger-scale advection played a minor role." However, many times later in the paper they attribute problem with the model-observation comparison to larger-scale issues (e.g., l.333, "...which could be caused by advection processes in reality modifying the residual layer"). A much stronger statemtent would look at many days when the surface forcing is dominant and then contrast this with many other days when the surface-forcing is not dominant. Then the authors could actually show the model does better (or worse) when surface forcing dominates rather than simply making a vague statements about it.

3. The smaller observed H and LE fluxes than modeled flux during the daytime (ie, Fig. 7) is almost certainly related to the choice of a 10-min averaging period to calculate the turbulent fluctuations. The authors acknowledge that there are low-frequency issues with the fluxes (ie, l.247-250) and during the daytime the time-scale involved for the

fluxes are longer than 10 minutes. Perhaps the Kaimal correction they describe fixes this issue, but using a 10-min window to calculate the fluxes is certainly a problem in the daytime (probably ok for nighttime). Why use a model to try and fix a methodology shortcoming? If a longer time window is used (e.g., either 30-min or an hour), it will make daytime obs H and LE larger and in closer agreement to the modeled fluxes.

4. p.3, l.71, why is the heat capacity assumed to be zero for vegetation-covered surfaces? Heat capacity has recently been shown to be an important consideration in land-surface models (e.g., Swenson, et al 2019). Getting the heat capacity of the storage terms (soil, biomass) correct is an important consideration to properly close the SEB (e.g., Lindroth, et al 2010, Leuning, et al 2012). These so-called "smaller" terms are important because they tend to have a phase shift (in terms of the diurnal cycle) relative to the other Rnet/H/LE flux terms. The authors appear to focus on the issue of low-frequency contributions to the fluxes, and do not talk about the heat storage terms as a possible problem (in fact, since heat capacity of the vegetation biomass is set to zero can the biomass storage term even be considered?).

5. The model has been designed to have many options and work with many different land-surface types (e.g., Table 1)...however, the evaluation is only done for one specific land-surface type. This is a very limited test of the validity of the model over the paramater space—I realize article length is an issue–but, what about evaluations of other surface types? At least maybe cover more than just one?

6. Though there is good information in Section 5.2, it seems like this section would benefit from subsections that guide the reader a bit better. As it is, I find it difficult to extract the key points the authors want to make from the comparison.

Specific Comments:
* * *
* does PALM stand for anything? Is this an acroynm? If so, it should be stated when

first introduced...

* p.4, l.98, remove parentheses with Duynkerke, 1999 reference

* p.6, l.159, is "high" vegetation, tall vegetation? such as trees?

* p.8, Table 1, why is C_0 set to 0.00 for all vegetation types?

* p.11, l.220, do waves have any effect on the transfer coefficients over water?

* p.12, l.244, "..but means of a Fourier extrapolation." Is there a reference for this method?

* p.12, l.244, Was a soil heat flux plate also used at some depth below or near the temperature measurements? If so, this is not clearly stated. Flux plates are typically used for measuring the soil flux while the soil temperature profile is used for the heat stored in the soil (e.g. see Eq. 7 and discussion in Leuning, et al 2012). [I now see this discussed on p.21, l.405-406]. Perhaps I don't fully understand this, but it seems like the comparison of the modeled and observed soil heat flux needs futher consideration. Are the same quantities actually being compared in Fig. 6?

* p.12, l.260, Fig. 2 is mentioned before Fig. 1.

* p.13, l.282, are the root fraction values based on measurements or assumed?

* p.16, Table 5, how were the specific values for each variable selected? For example, LAI has values of 0.5 to 3 m2/m2. Are these realistic or reasonable values? Furthermore, if you want to truly look at the sensitivity to LAI (or other variables), why not vary them between the endpoints, e.g., in steps of 0.1 m2/m2 between 0.5 and 3?

* p.18, l.345-374, I understand there is a difference in Rnet which is presumably due to an incorrect modeled surface temperature. But, I'm not sure what to take away from the discussion following this—is the suggestion that the LAI should really be 0.5 m2/m2? Is the problem with the observations since the radiative flux divergence is not included?

* p.21, l.410, "It strikes", should be "It is striking"?

* p.21, l.413-416, are you suggesting that the observed LAI is incorrect? If you increase LAI, LE should increase at the expense of H..this is not surprising.

* p.24, l.472, for more info on grid spacing of models in stable conditions, see Sullivan, et al 2016.

* p.28, l.575, "differences of up to 50% are possible.". Differences in which variable?

* p.29, l.602-603, I didn't see how step-like orography is implemented in the LSM? Was this described somewhere in the paper?

References:

————

Leuning, R., van Gorsel, E., Massman, W. J., and Isaac, P. R., 2012: Reflections on the surface energy imbalance problem, Agricultural and Forest Meteorology, 156, 65-74, doi:10.1016/j.agrformet.2011.12.002

Lindroth, A., Molder, M., and Lagergren, F., 2010: Heat storage in forest biomass improves energy balance closure, Biogeosciences, 7, 301-313, doi:10.5194/bg-7-301-2010

Swenson, S. C., Burns, S.P. , and D.M. Lawrence, 2019: The impact of biomass heat storage on the canopy energy balance and atmospheric stability in the Community Land Model. Journal of Advances in Modeling Earth Systems (JAMES), 11, 83-98, doi:10.1029/2018MS001476

Rodell, M., P. R. Houser, A. A. Berg, and J. S. Famiglietti, 2005: Evaluation of 10 Methods for Initializing a Land Surface Model. J. Hydrometeor., 6, 146-155, https://doi.org/10.1175/JHM414.1

Sullivan, P.P., Weil, J.C., Patton, E.G., Jonker, H.J.J., Mironov, D.V., 2016: Turbulent winds and temperature fronts in large-eddy simulations of the stable atmospheric boundary layer. J. Atmos. Sci. 73, 1815-1840. doi:10.1175/JAS-D-15-0339.1

---

## Author Comment (AC1) · 1 Feb 2021

We would like to thank the anonymous referee for the time and effort that was devoted to reviewing the paper. The original manuscript was suffering from technical flaws and could be significantly improved upon.

Reviewer Comment (RC): Within this study, the authors introduce the land-surface model (LSM) implemented in PALM model, and evaluate the performance using two-day in-situ observations. They conduct a series of sensitivity experiments to explore

the impacts of model parameters on simulating the boundary-layer profiles, the surface energy balance, and nearsurface meteorological variables. Despite the detailed description about the LSM and useful information for PALM users, the results are very preliminary. As the manuscript reads now, the authors touched some subjects only briefly without really adding any scientific merit. It is more like a graduate's project essay than a scientific paper.

Author's reply (AR): In principle, the manuscript is a technical model description paper (which fits nicely in GMD in that aspect). The study and sensitivity experiments serve both to provide a first evaluation and give an idea how sensitive results are to the selection of various parameters of the model. With that in mind, we do not agree that there is no scientific merit as the manuscript was not aiming at communicating any new scientific findings (and that is also not was we would expect in a model description / validation paper). Being authors of (together) more than 40 peer-reviewed papers, we are fairly sure that the quality of the manuscript is adequate for a scientific paper.

\*\*Specific comments:\*\*

RC: The manuscript needs significant restructuring in order to establish a better focus within this paper. I suggest the authors seriously consider the sensitivity experiment design first. Of course every parameter in a model would have more or less impacts on the simulations. To fit this study, no need to take the radiation scheme (RRTMG & CLEARSKY), large-scale forcing (ADV_tq), or resolution (Dz_2) into account. Other case like EMIS_95 and EMIS_100, even not being mentioned in the manuscript. These unnecessary results make the paper more difficult to follow and do not have much scientific merit being covered so briefly. Removing the relevant content would be better, in my opinion.

AR: Modelers often face a situation in which they want to model a real case scenario but not all input parameters of the site are known. With this in mind, the sensitivity experiment was designed. In this respect, the choice of the radiation scheme might

be important. Case ADV_tq is needed to explain a mismatch between simulations and observation in the total water mixing ratio (L528-529 of initial manuscript). Case Dz_2 is used to discuss a possible explanation of the mismatch of the nocturnal boundary layer (L472-476 of initial manuscript). In our opinion, it is justified to address the mentioned 4 cases, which are not directly linked to the land surface. The other 17 cases are selected as endpoints of realistic values for each parameter. It is a very valid point that cases EMIS_95 and EMIS_100 were not mentioned in the original manuscript. In fact, both cases are always among the gray lines in Figs. 3-6, however they do not influence the energy balance components. We have added this to the discussion of the relevant section.

RC: Second, the results section seems poorly phrased. I see a little conjecture and repetition in Section 5. For example in L430-445, this portion could be removed (at least be shortened), as it does not provide much "facts" to convince readers. If I correctly understand, the point is the observed H and LE might be underestimated due to the limitation of eddy-covariance method, which partially explains the overestimated H and LE by model. Fig. 7 can be removed as well because we've already got those information from Figs. 3-6. The black line of RES term just indicates the measurements are of bad quality. Plus, a repeated statement about Bowen ratio in the end; authors have mentioned that in L396-400. For L472-489, after going through this paragraph, I still have no idea why the model is not able to reproduce the nocturnal boundary layer, even feel a big unsolved issue existed in the LSM or the atmospheric model. Authors should not do like give a hypothesis, reject it, and then say we in actual don't have much confidence in the rejection. This discussion won't help raise one's interest in the model.

AR: We acknowledge that the information shown in Fig. 7 can be deduced from Figs. 3-6, nevertheless the reader benefits from this figure, because it points out an important issue of simulation -observation comparisons: which is correct, the model or the measurement? With this figure we intend to emphasize this important issue, and further we intend to give a rough measure of the uncertainty which comes along with such a model-observation comparison. The authors have high confidence in the observation dataset of CESAR. "The black line of RES term" rather indicates the energy-balance-closure problem, which shows that about 25% of the available energy is missing. Similar or even higher residuals can be found in all flux observations. We agree that, to a certain degree, it was difficult to identify the key points of the results in the original manuscript. To better guide the reader, subsections have been added to section 5. Paragraphs have been restructured and a sentence about the Bowen ratio has been deleted to avoid repetition. Regarding the issue to simulate the nocturnal boundary layer, we need to stress that this is a common finding for atmospheric models (see e.g., van Stratum, B. J. H. and Stevens, B.: The influence of misrepresenting the nocturnal boundary layer on idealized daytime convection in large-eddy simulation, Journal of Advances in Modeling Earth Systems, 7, 423–436, https://doi.org/10.1002/2014MS000370, 2015). There is abundant other literature on this issue and not related to the LSM or the particular LES model in use. One of the main problems with the nocturnal boundary is the representation of the dominant turbulent eddies. As turbulence is damped during nighttime by stratification, the dominant scales are much smaller than during daytime. As a consequence, a smaller grid spacing is usually required. In the scope of the present work, it was not possible to run the simulations at much higher resolution, so that we might ascribe some effects during nighttime to the coarse resolution.

RC: The third point is to be correct. Like L355, I assume ALBE_24 is the sensitivity experiment featuring a decreased albedo in comparison to the REF. But following L232, the shortwave albedo is set to 0.14 in REF. Please double check the setup of your experiments. L398: I doubt the discussion "cases HUMID_sat and LAI_05 show significantly lower Bowen ratios compared to observations". From Figs 4&5, I see larger H and lower LE which means a larger Bowen ratio (H/LE) than observations. In L509, it is the low temperature leading to stable boundary layer, not "stable layer, hence the low temperature". Likewise later in L513, the convective boundary layer started developing because of the surface heating in the morning than "the stable layer is eroded and temperature can rapidly increase".

AR: Thank you for spotting these errors, we have corrected them accordigly. Regarding the albedo sensitivity experiment, the naming was misleading and ist now consistently based on the shortwave albedo (was longwave albedo before).

RC: Lastly, seriously improve the English writing.

AR: The manuscript was now proof-read by a native speaker.

\*\*Technical comments\*\*

RC: L1: PALM is an acronym?

AR: Even though the PALM developers do not want to use the long name (abbreviation for Parallelized Large-eddy Simulation Model) anymore and this paper is part of a special issue featuring PALM, we have included a note in the revised manuscript, because readers, who are unfamiliar with the model expect some kind of explanation.

RC: L2: "For this" -> "To this end"

AR: As suggested by a native speaker, we removed this phrase.

RC: L4: Add "with observations" after "agree well"

AR: Done.

RC: L8 & L47: "By this" -> "In this way"

AR: We removed this phrase.

RC: L235: What is CESAR?

AR: Cabauw Experimental Site for Atmospheric Research (CESAR) - was added.

RC: L263-264: Rephrase the sentence to "The CESAR site is well equipped with the vegetation and soil information which provides a good opportunity to evaluate the land-

surface parameterization proposed in the present study."

AR: Thank you for this suggestion, we rephrased the sentence accordingly.

RC: L267-271: Change to "The land surface scheme configuration is given in Table 4" and then add the information you don't have in Table 4.

AR: Redundant information was removed from the text.

RC: L314: "One the one hand" -> "On one hand"

AR: We removed this phrase.

RC: Fig.2: Crowded figure. May be plotted as Fig. 9, one time in one panel.

AR: In the revised manuscript, the profiles are depicted in two separate figures with only one day plotted at a time.

RC: L326, L334 & L337: Add "with observations" after "agree well"

AR: Done.

RC: L377: "Moreover, the simulated H ..., respectively" -> "The model overestimates H

AR: Thank you.

---

## Author Comment (AC2) · 1 Feb 2021

Summary:

Reviewer Comment (RC): This paper describes the land-surface model in the PALM model system. The subject matter is approprite for the GMD journal and the text is well written. The details about the model are provided in appropriate amount of detail, equations provided are also appropriate. As part of the paper, there is a "first evalua-tion" which I think could/should be improved upon, as detailed in my comments below.

[Figure]
Even though this is a "first evaluation" I think it can be done with more rigor and there are, in my opinion, a few technical flaws. While I think the description of the model is ready to be published, the evaluation of the model (and comparison with observations) needs improvement before that portion of the paper should be published.

Author's reply (AR): In the name of all athors, I would like to thank the anonymous referee for the time and effort that was devoted to reviewing the paper. The valuable comments helped to significantly improve the original manuscript.

General Comments:

RC: 1. Most land-surface models vs observation comparisons I am aware of use the model run in "single-point" mode with the observed tower data driving the model. However, the model output from the PALM LSM has been averaged over the spatial domain. This spatial seems to confound the comparison (e.g., l.337-340). Is there a reason the spatial averaging of the model data is necessary?

AR: In (LES) modeling, it is common practice to employ the spatial average instead of a temporal average as it is done for single-point observations. It would be possible to mimic observations by using a time-averaging window to remove/reduce the turbulent signal from the LES data. Nonetheless, accodring to Taylor's Hypothesis, a spatial signal in our LES model should relate to a temporal signal in the observations as long as the surface is homogeneous. From a practical point of view, using the spatial average at one point in time is rather convenient and requires much less memory (it can be calculated on-the-fly during the simulation). Physically, the spatial average is in general superior over a temporal average, because Taylor's hypothesis is affected by changes of the mean wind direction, non-stationarity of the flow, and self-correlation. Because a spatial average over homogenous terrain is generally equivalent to the single-point temporal average we decided to use the standard output of PALM (which is the horizontally-averaged data). This issue is not present in RANS simulations with horizontally homogeneous surface, as the RANS model only provides the

time-averaged flow so that there are no spatial and temporal variations due to turbulence. In order to add information about the spatial variability of the shown LES data, we have included the range (minimum to maximum value) in Figs. 3-6.

RC: 2. The 2-day period seems too short to do a thorough evaluation of the model. There are many decades of Cabauw data, but only a short 2-day period is used. Even for a "first evaluation" this seems like a weakness of the paper. According to the authors, this particular period was chosen was because (l.236) "...the forcing from the surface was dominant and larger-scale advection played a minor role." However, many times later in the paper they attribute problem with the model-observation comparison to largerscale issues (e.g., l.333, "...which could be caused by advection processes in reality modifying the residual layer"). A much stronger statemtent would look at many days when the surface forcing is dominant and then contrast this with many other days when the surface-forcing is not dominant. Then the authors could actually show the model does better (or worse) when surface forcing dominates rather than simply making a vague statements about it.

AR: Even though the Cabauw site is one of Europe's most frequently used sites for this kind of comparison, it is not easy to find a period where there is little advection, no clouds and all measuring instruments are running. The period we chose was suggested by the person in charge for the Cabauw site (Fred Bosveld), who has great experience in the data quality and availability. A major reason for this particular period was that in May 2008 radiosondes were launched three times daily as part of the IMPACT-EUCAARI campaign (Kulmala et al., 2009), which we used for initialization. Overall, the period in question did have a few more days, but as we would have had to incorporate some kind of nudging to the forcing and/or data assimilation to avoid model drift, we decided to simulate a two-day period only. This gives the model the possibility to study the behavior of the model over a full diurnal cycle and also look at how the first day affects the following day(s). In general, it is not possible the derive the height-dependent boundary layer advection from observation needed to drive the

LES model, particularly within the boundary layer, where turbulence dominates, such large-scale tendencies are difficult or impossible to obtain. We thus think that for a "first evaluation" the chosen period is acceptable. We have added a brief discussion about this to the manuscript which points out why we have chosen exactly this period.

RC: 3. The smaller observed H and LE fluxes than modeled flux during the daytime (ie, Fig. 7) is almost certainly related to the choice of a 10-min averaging period to calculate the turbulent fluctuations. The authors acknowledge that there are low-frequency issues with the fluxes (ie, l.247-250) and during the daytime the time-scale involved for the fluxes are longer than 10 minutes. Perhaps the Kaimal correction they describe fixes this issue, but using a 10-min window to calculate the fluxes is certainly a problem in the daytime (probably ok for nighttime). Why use a model to try and fix a methodology shortcoming? If a longer time window is used (e.g., either 30-min or an hour), it will make daytime obs H and LE larger and in closer agreement to the modeled fluxes.

AR: We agree with the reviewer that 10-minute intervals averaging time during daytime are certainly too short and the natural choice would be 30 minutes, though even 30-minute intervals are often not sufficient under convective conditions. For the current Cabauw data, 10 minute intervals are used and processed/corrected according to the Cabauw standard. Unfortunately we do not have access to the raw data (i.e. the raw data is not stored permenantly) to calculate fluxed based on a longer time window. However we discuss the shortcomings of the method in the manuscript.

RC: 4. p.3, l.71, why is the heat capacity assumed to be zero for vegetation-covered surfaces? Heat capacity has recently been shown to be an important consideration in land-surface models (e.g., Swenson, et al 2019). Getting the heat capacity of the storage terms (soil, biomass) correct is an important consideration to properly close the SEB (e.g., Lindroth, et al 2010, Leuning, et al 2012). These so-called "smaller" terms are important because they tend to have a phase shift (in terms of the diurnal cycle) relative to the other Rnet/H/LE flux terms. The authors appear to focus on the issue of low-frequency contributions to the fluxes, and do not talk about the heat storage terms

as a possible problem (in fact, since heat capacity of the vegetation biomass is set to zero can the biomass storage term even be considered?).

AR: The implementation of vegetated surfaces in PALM is based on the parameterization used in the LSM of the Integrated Forecast System (IFS). Accordingly, the skin layer has no heat capacity (IFS Documentation). This is because the heat storage of the vegetation layer is difficult to estimate and would introduce another parameter of uncertainty. Please also note that the vegetation in the simulated domain is short grass (homogeneous), whereas Swenson, et al 2019 and Lindroth, et al 2010 study heat stored in forests, where the heat capacity of the canopy is much much higher than for short grass. The assumption to have zero heat capacity for short grass is not unrealistic and thus a valid approach (we have added a brief discussion about this to the manuscript). The heat capacity of the soil is treated properly by the soil model.

RC: 5. The model has been designed to have many options and work with many different land-surface types (e.g., Table 1)...however, the evaluation is only done for one specific land-surface type. This is a very limited test of the validity of the model over the paramater space—I realize article length is an issue–but, what about evaluations of other surface types? At least maybe cover more than just one?

AR: The evaluation is done for 21 cases of which 11 affect land-surface parameters (ALBE_24, ALBE_44, CAP_2e4, COND_2, COND_6, EMIS_95, EMIS_100, LAI_05, LAI_3, ROUGH_01, ROUGH_001), and thereby the land-surface type "short grass" is implicitly altered. In our opinion, the parameter space thus appears to be systematic. We acknowledge that adding some more of the pre-set land-surface types (Table 1) may be in the interest of the user, however, this requires more observational sites over the respective homogeneous surface types, which are not easily available with similar data quality as of the Cabauw site. In particular, it is difficult to find locations that are horizontally homogenous. This becomes particularly true for artifical surfaces such as pavements. Nevertheless, we are planning to evaluate the land surface model also for pavements based on observational near-surface data obtained during a measurement

campaign on a disued movement area of an airport in Berlin, Germany.

RC: 6. Though there is good information in Section 5.2, it seems like this section would benefit from subsections that guide the reader a bit better. As it is, I find it difficult to extract the key points the authors want to make from the comparison.

AR: Thank you, we have included subsections accordingly and restructured the text in some parts to better point-out the key points.

Specific Comments:

* RC: does PALM stand for anything? Is this an acroynm? If so, it should be stated when first introduced...

AR: Even though the PALM developers do not want to use the long name (abbreviation for Parallelized Large-eddy Simulation Model) anymore and this paper is part of a special issue featuring PALM, we have included a note in the revised manuscript, because readers, who are unfamiliar with the model are expecting some kind of explanation.

* RC: p.4, l.98, remove parentheses with Duynkerke, 1999 reference

AR: Thank you for spotting this.

* RC: p.6, l.159, is "high" vegetation, tall vegetation? such as trees?

AR: Yes, we changed it accordingly.

* RC: p.8, Table 1, why is C_0 set to 0.00 for all vegetation types?

AR: The surface heat capacity is set to 0 by default, because this is, how it is done in the IFS code. Even though this is definately unsuitable for tall vegetation such as forests, we do not give explicit values, because, to meet our standard, they must be comprehensively tested to not mislead the user. Nevertheless, the user has the possibility to change the value. We added a note to the revised manuscript that this value should be carefully adjusted, if e.g. a forest is simulated.

* RC: p.11, l.220, do waves have any effect on the transfer coefficients over water?

AR: In the case of inland water, say for small lakes, rivers, ponds, etc., the transfer coefficients are not altered. For ocean, the roughness lengths vary to account for sub-grid scale waves. Here we use a Charnock parameterization. This information was given in line 94-96. We added another note to the section about treatment of the water surfaces to improve readability.

* RC: p.12, l.244, "..but means of a Fourier extrapolation." Is there a reference for this method?

AR: A reference (Bosveld, 2020) was added to the revised version.

* RC: p.12, l.244, Was a soil heat flux plate also used at some depth below or near the temperature measurements? If so, this is not clearly stated. Flux plates are typically used for measuring the soil flux while the soil temperature profile is used for the heat stored in the soil (e.g. see Eq. 7 and discussion in Leuning, et al 2012). [I now see this discussed on p.21, l.405-406]. Perhaps I don't fully understand this, but it seems like the comparison of the modeled and observed soil heat flux needs futher consideration. Are the same quantities actually being compared in Fig. 6?

AR: Surface soil heat flux can also be derived from the soil heat flux observations alone. We resolve the 24h time series of G05 and G10 in its Fourier components. Corresponding components can then be extrapolated to the surface in the same way as described above for the diurnal Fourier component. Subsequently an inverse Fourier transformation is performed on the extrapolated components to construct the time series of the surface soil heat flux. The penetration depth for short time scales becomes small. This means that for these high frequency components the signal may be hidden in distortions of the observations, either noise or deviations because the time series is not a response to a perfect cyclic forcing with a period of 24 hours. In the current implementation the first 9 Fourier components are used, thus the fastest resolved cycle has a length of 2h40m. Extrapolation of component is done when the amplitude of the

10 cm Fourier component is > 1 W m-2 and when the amplitude of the 5cm sensor is less than 3 times the amplitude of the 10 cm sensor. If this conditions are not met the amplitude of the 5 cm sensor is used.

* RC: p.12, l.260, Fig. 2 is mentioned before Fig. 1.

AR: We established the correct order.

* RC: p.13, l.282, are the root fraction values based on measurements or assumed?

AR: Root density is based on the study of Jager (1976) and assumed for model layers that are in-between the observed layers.We clarified this accordingly.

* RC: p.16, Table 5, how were the specific values for each variable selected? For example, LAI has values of 0.5 to 3 m2/m2. Are these realistic or reasonable values? Furthermore, if you want to truly look at the sensitivity to LAI (or other variables), why not vary them between the endpoints, e.g., in steps of 0.1 m2/m2 between 0.5 and 3?

AR: In the revised manuscript we have specified our choices. Indeed, starting from the reference case which was setup as a best guess based on reports of the Cabauw site, we have varied the respective parameter in a reasonable range. We agree with the reviewer that we do not show a full parameter study but rather a small part of the parameter range, which we now make more clear. With this study we did not intend to provide a comphrehensive parameter study but an idea on how sensitive the model reacts on specific parameters. This is mainly motivated by the fact that in many cases these input data is either not available and need to be estimated, or often only roughly available. In both cases the uncertainty in the estimated parameters is high. We now try to make this more clear at the end of the setup description.

* RC: p.18, l.345-374, I understand there is a difference in Rnet which is presumably due to an incorrect modeled surface temperature. But, I'm not sure what to take away from the discussion following this—is the suggestion that the LAI should really be 0.5 m2/m2? Is the problem with the observations since the radiative flux divergence is not

included?

AR: The paragraph provides a discussion on how sensitive the surface net radiation reacts on changes of specific model parameters. We do not intend to give the impression that there is one truth and one perfect parameter combination, so thanks for pointing this out. Instead our intention is to outline the sensitivity of the energy balance components, here surface net radiation, on specific parameter variations. In the revised manuscript we try to make this more clear what the intention is.

* RC: p.21, l.410, "It strikes", should be "It is striking"?

AR: Thanks for this hint. In the revised version we have changed it to "remarkably".

* RC: p.21, l.413-416, are you suggesting that the observed LAI is incorrect? If you increase LAI, LE should increase at the expense of H..this is not surprising.

AR: Removed sentence to avoid confusion.

* RC: p.24, l.472, for more info on grid spacing of models in stable conditions, see Sullivan, et al 2016.

AR: Thank you, the reference has been added.

* RC: p.28, l.575, "differences of up to 50% are possible.". Differences in which variable?

AR: H and LE. The sentence has been rearranged for clarification.

* RC: p.29, l.602-603, I didn't see how step-like orography is implemented in the LSM? Was this described somewhere in the paper?

AR: The reviewer is right, these information was given out of the context. We have now outlined the issue to make our point more clear.

References:

Bosveld, F.: The Cabauw In-situ Observational Program 2000 – Present: Instruments,

Calibrations and Set-up, Tech. report 384, KNMI, De Bilt, The Netherlands, regularly updated, 2020. IFS Documentation, Physical Processes, Chapter 8 Surface parameterisation, 2016, http://www.ecmwf.int/en/elibrary/16648-part-iv-physical-processes

Jager, C., Nakken, C., and Palland, C.: Bodemkundig Onderzoek van twee Grasland-percelen Nabij Cabauw, NV Heidemaatschappij Beheer, in Dutch, 1976.

Kulmala, M., Asmi, A., Lappalainen, H. K., Carslaw, K. S., Pöschl, U., Baltensperger, U., Hov, Ø., Brenquier, J.-L., Pandis, S. N., Facchini, 775 M. C., Hansson, H.-C., Wiedensohler, A., and O'Dowd, C. D.: Introduction: European Integrated Project on Aerosol Cloud Climate and Air Quality interactions (EUCAARI) – integrating aerosol research from nano to global scales, Atmospheric Chemistry and Physics, 9, 2825–2841, https://doi.org/10.5194/acp-9-2825-2009, 2009

Lindroth, A., Molder, M., and Lagergren, F., 2010: Heat storage in forest biomass improves energy balance closure, Biogeosciences, 7, 301-313, doi:10.5194/bg-7-301-2010

Swenson, S. C., Burns, S.P. , and D.M. Lawrence, 2019: The impact of biomass heat storage on the canopy energy balance and atmospheric stability in the Community Land Model. Journal of Advances in Modeling Earth Systems (JAMES), 11, 83-98, doi:10.1029/2018MS001476

---

## Author Response (AR2)

Comments to the Author:

One of the original reviewers has considered the revised manuscript. That reviewer had one minor comment that they would like to see addressed.

L508: Clarify at the start of this paragraph that the misrepresentation of the nocturnal boundary layer is a common issue for atmospheric models. Due to the smaller turbulence scales at nighttime than at daytime, a smaller grid spacing is usually required, as clarified in the response to my original review. And that gives authors the intuition that the REF grid spacing may be too coarse.

The comment is addressed at the beginning of the paragraph in the revised manuscript.

(L508-510)

Please address this comment in a revised manuscript and, if adequately addressed, this paper will be suitable for publication.

The marked-up manuscript version showing the changes highlights many numbers, because they were put into math mode. Furthermore, the sections "code and data availability" as well as "author contributions" have been revised, which is not highlighted by latexdiff.